# For Wastewater matters: Incorporating wastewater treatment and reuse into a process-based hydrological model (CWatM v1.08)

Journal: Geoscientific Model Development Discussions

Dor Fridman[1], Mikhail Smilovic[1], Peter Burek[1], Sylvia Tramberend[1], Taher Kahil[1]

[1] Water Security Research Group, Biodiversity and Natural Resources Program, International Institute for Applied Systems Analysis (IIASA), Laxenburg, Austria.

*Correspondence to:* Dor Fridman (fridman@iiasa.ac.at)

**Abstract**

Wastewater treatment and reuse are becoming increasingly critical for enhancing water use efficiency and ensuring reliable water availability. Wastewater also significantly influences hydrological dynamics within urban watersheds. Although hydrological modeling has advanced to incorporate human-water interactions, large-scale and multi-resolution models often lack comprehensive integration of wastewater treatment and reuse processes. This paper presents the new wastewater treatment and reuse module as part of the hydrological Community Water Model (CWatM) and demonstrates its capabilities and advantages in an urban watershed with intermittent flows. Incorporating wastewater into the model improves model performance by better representing low- and peak-flows during the dry and wet seasons. It allows for representing wastewater reuse in different sectors and exploring different measures for increasing wastewater reuse, and its effects on the water stress level. Modeling of wastewater treatment and reuse is relevant for many regions around the world with similar climates or urbanization patterns, or those promoting wastewater reuse. The wastewater treatment and reuse module could be upscaled by minimizing the data requirments through a simplified workflows. Combined with the availability of recent datasets of wastewater treatment plants and processes, a global application of the module is feasible. As the current development focuses on water quantity, the water quality dimension of wastewater treatment remains a limitation, which sets the plans of incorporating water quality into the model and developing global input data for wastewater treatment and reuse.

## 1. Introduction

Hydrological modeling has developed over the last few decades to account for the human-water interface (Wada et al., 2017). Recent developments in this field focused on developing higher-resolution global hydrological models (GHMs) by increasing models' spatial resolution, adjusting their datasets, and including a variety of water management options (Abeshu et al., 2023; Hoch et al., 2023; Burek et al., 2020; Hanasaki et al., 2022).

Increasing human interventions in the water cycle and higher spatial resolution modeling have emphasized the need to include water management as an integral part of hydrological models (Hanasaki et al., 2022). Some large-scale hydrological models (LHMs) already account for water management aspects, like water withdrawal and consumption, irrigation management, reservoir operations, water transfers, and desalination (Wada et al., 2017). Wastewater treatment and reuse are other management options that are increasingly important in many regions. Currently, treated wastewater is estimated at 188 $km^3$ per year globally, which is around 52% of effluents generated. Further, approximately 22% (of treated wastewater) is estimated to be reused(Jones, van Vliet, Qadir, & Bierkens, 2021). Thebo et al. (2017) find that around 35.9 mega hectares of irrigated cropland are supported by rivers dominated by wastewater from upstream urban areas, and Van Vliet et al. (2021) indicate that expansion of treated wastewater uses from 1.6 to 4.0 billion $m^3$ per month can strongly reduce water scarcity levels worldwide.

Specifically, wastewater reuse is a valuable water source for industrial use and irrigation in water-stressed regions. For example, Israel reuses around 88% of its treated wastewater, mainly for use in the agricultural sector, where it satisfies about 45% of the agricultural water withdrawals (Fridman, Biran, & Kissinger, 2021). Treated wastewater is also used for irrigation in South European, Mediterranean, and North African countries (Angelakis et al., 1999; Bixio et al., 2006). While accepting exacerbated stress on freshwater resources, the European Parliament is working to improve the quality of wastewater treatment in the EU, aiming to increase wastewater reuse (European Parliament, 2024). It follows that prospects of increased utilization of this resource are plausible. Wastewater collection, treatment, and reuse are relevant processes for the hydrological modeling of urban catchments and complex water resource systems and are included in different small-scale models (Salvadore, Bronders, & Batelaan, 2015). Large-scale hydrological models often neglect wastewater treatment and reuse. However, to some extent, few models include wastewater treatment effects on water quality. The Soil & Water Assessment Tool (SWAT) includes septic tanks as an on-site treatment option. It simulates the percolation of wastewater into soils, the interaction between pollutants and the soil media, and bacteria build-up and nutrient uptake (Neitsch, Arnold, Kiniry, & Williams, 2011).

Another example is DynQual, a global water quality model coupled with the PCR-GLOBWB2 hydrological model (Jones et al., 2023). The model includes wastewater treatment processes in water quality simulations while simplifying wastewater treatment and reuse management. Namely, in DynQual, wastewater is generated, collected, treated, and discharged locally (in a single grid cell).

While these are significant developments, they only partially capture the complex dynamics between human activities and hydrological processes occurring in urbanized catchments or otherwise complex water resource systems.

This paper introduces a recently developed, customizable wastewater treatment and reuse module as part of the Community Water Model (CWatM), allowing various modes of simulating wastewater treatment and reuse processes.

CWatM is a versatile, fully distributed, modular, and open-source hydrological model that simulates natural and
human-affected hydrological processes at a daily time step and multiple spatial resolutions ranging from 0.5° to
30 arc-seconds (Burek et al., 2020). CWatM has extensive and publicly available documentation of the source
code, the model structure, and model training and tutorials (https://cwatm.iiasa.ac.at/, last access: July 11th, 2024).
The development of the wastewater treatment and reuse fits with the modularity and flexibility of CWatM by
providing various modus-operandi to enable simulation of wastewater treatment and reuse on global (0.5°),
regional (5 arc minutes), and local (up to 30 arc seconds) scales. This paper aims to introduce this module using
a high-resolution (around 1 km$^2$) case study of an urbanized river basin in a relatively dry climate (the Ayalon
River basin in Israel).
The rest of the paper is organized as follows. Section 2 describes the model development; section 3 covers the
case study, input data, and scenarios; and section 4 presents the results, followed by discussion and conclusions
in sections 5 and 6, respectively.

## 80     2. Module development and description

### 81     2.1. The Community Water Model (CWatM)

CWatM is a large-scale distributed hydrological model suitable for implementation at global and regional scales
(Burek et al., 2020). It is implemented in the Python programming language and is fully open-source
(https://cwatm.iiasa.ac.at). CWatM simulates the main hydrological processes and covers some aspects of the
human-water interface. This paper presents the recently developed wastewater treatment module to enhance
CWatM's capacities for addressing human water management. The model is applied to the relatively water-scarce
Ayalon River basin in Israel. It uses a spatial resolution of 30 arc seconds (~1 km$^2$ grid) in a geographic coordinate
system (WGS84). Groundwater is simulated by the coupled CWatM-MODFLOW6 model (Guillaumot et al.,
2022) at a spatial resolution of 500 meters using the UTM36N coordinate system.

### 90     2.2. Developing the Wastewater Treatment and Reuse Module (WTRM)

The wastewater treatment and reuse module (WTRM) enhances the capacity of CWatM to simulate the human-
water interface at high spatial resolution. It introduces wastewater collection, treatment, disposal, and reuse to
CWatM. Large-scale modeling shall utilize the basic setup of the WTRM for which sufficient data is available
globally. Case studies with higher data availability may benefit from optional advanced functions. The following
section distinguishes between basic and advanced (optional) model processes. Figure 1(A) demonstrates the
WTRM workflow, split into three sub-processes: (1) pre-treatment, (2) treatment, and (3) post-treatment, and
differentiates between the CWatM existing (gray boxes) and newly added features (green boxes).

**(A)**

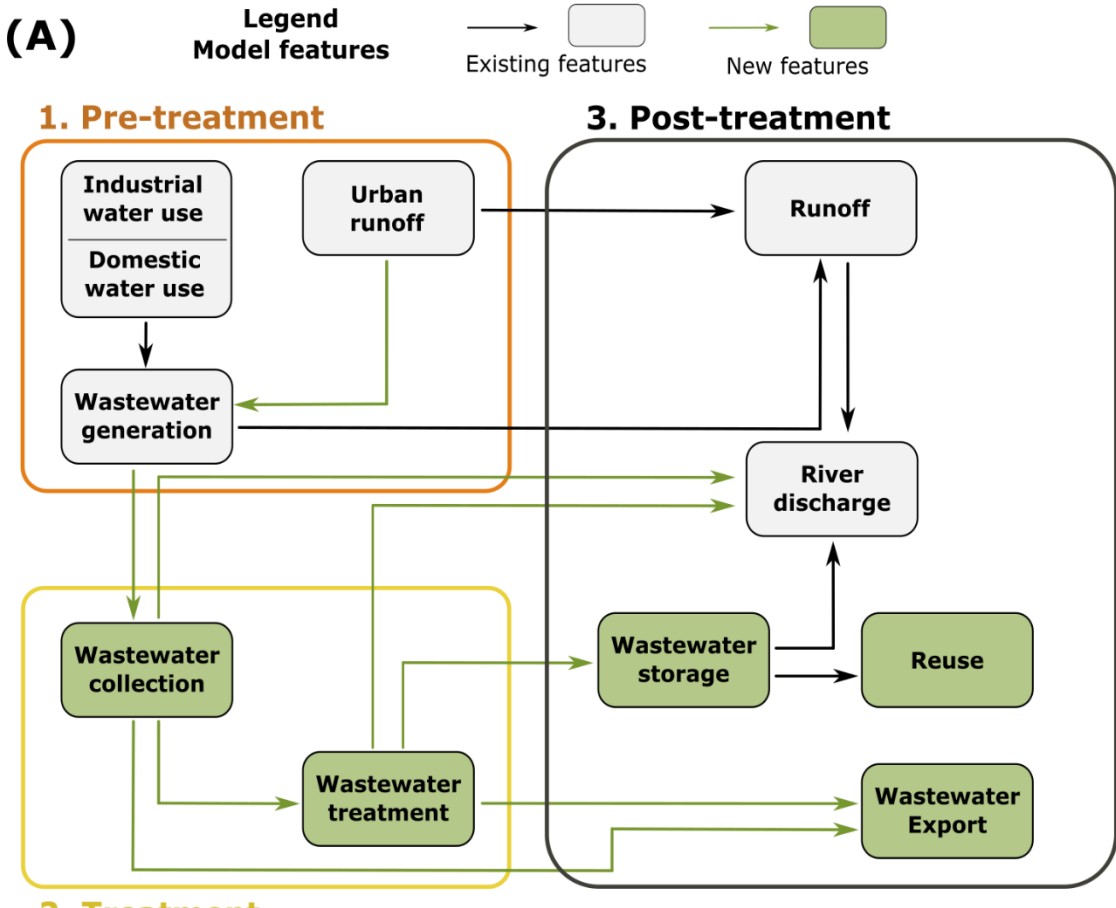

**(B)**

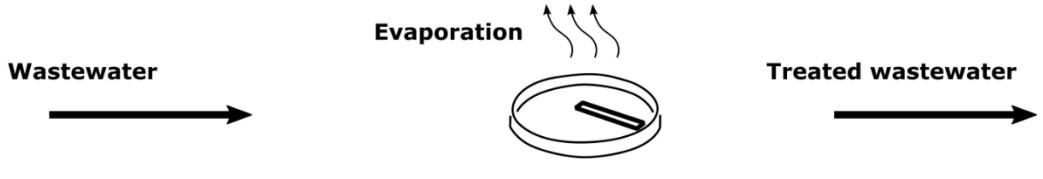

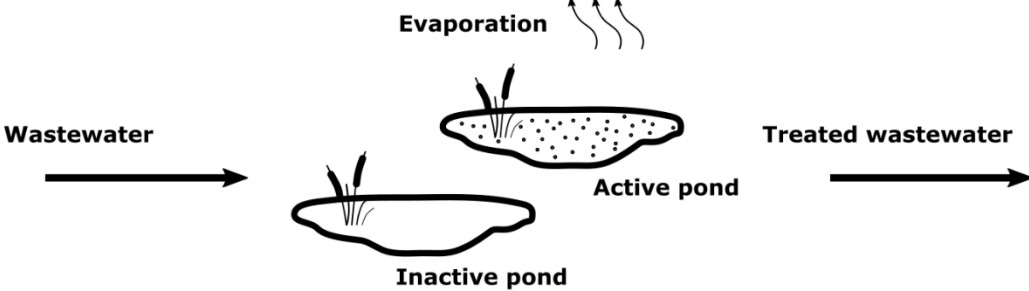


**Figure 1: (A) The WTRM new features (green boxes and arrows) and their interactions with the existing feastures of CWatM (gray boxes and arrows), and (B) Water balance for the intensive and extensive wastewater treatment systems.**

### 2.2.1. Pre-treatment: wastewater generation and collection

Wastewater generation in CWatM is represented by non-irrigation return flows, which are a function of water availability and sectoral allocation scheme, and the ratio between the consumptive and total water withdrawal. The wastewater module estimates domestic and industrial wastewater generation ($Eff^{Dom}$ and $Eff^{Ind}$) by multiplying the non-irrigation return flows by the relative sectoral water demand. The next step is to collect and supply wastewater to wastewater treatment plants (WWTP) (see Equation 1).

**Equation 1: Calculating WWTP influents in WTRM.**

$$Inflow_{j,t} = \sum_{l \in j} \left( Eff^{Dom}_{l,t} \times D^{Dom}_j + Eff^{Ind}_{l,t} \times D^{Ind}_j \right) \times Cs_l + Rf_l \times \alpha$$

*Note: j* and *t* represent a simulated WWTP and the time step, respectively; *l* indicates a grid cell. Table 4 describes all the WTRM variables, data sources, and default values.

WWTP service areas (or collection areas) are model input that defines the linkages between the location of wastewater generation (individual grid cells, denoted by *l*) and wastewater treatment plants (denoted by *j*), namely that the wastewater from all grid cells in a collection area are treated in the associated WWTP (see Figure S10) Wastewater collection is also a function of the sewer connection rate ($Cs_l$), where a value of one indicates all wastewater is collected and sent to a WWTP. Moreover, it can include urban runoff ($Rf_l$) due to leakage or integration of the urban stormwater and wastewater systems. The $\alpha$ coefficient defines the system integration level and ranges from zero (no integration) to one (complete systems integration). The total wastewater collected in all grid cells *l* associated with a WWTP *j* is registered as the treatment plant's inflow.

Modeling sector-specific WWTP (e.g., treatment of only industrial wastewater) is an advanced model functionality, and to-date does not fit a global application. It uses a boolean variable (e.g., $D^{Dom}$), which equales one if the treatment plant receives a specific wastewater stream (e.g., domestic). A default value of one for both sectors is set in place in case of missing data.

### 2.2.2. Treatment: Influent, evaporation, and effluent

Simulated wastewater treatment plants must have the following basic features: location, start year of operation, daily treatment capacity, treatment period (days), and outflow location.

Currently, the module supports two optional wastewater treatment technologies associated with the treatment period. The two options are intensive and extensive treatment plants, as described in Figure 1(B) 4 and 5. Intensive treatment refers to the conventional wastewater treatment technology characterized by low residence time and low area requirements. It treats water to secondary or tertiary levels over less than 24 hours (Pescod, 1992). CWatM uses a daily timestep, so the intensive treatment plant's treatment period is set to one day. Any WWTP with a longer treatment period (i.e., >= 2 days) would be classified as extensive. Extensive treatment refers to natural biological systems, consisting of a short primary treatment in a relatively deep anaerobic pond, followed by a longer residence time (20-40 days) in a shallow facultative pond for secondary treatment (Pescod, 1992). An advanced model feature enables the exceedance of the WWTP daily capacity by temporarily reducing the hydrological retention time (HRT). This feature is enabled by setting a treatment plant-specific minimally allowed HRT, providing WWTP some buffer to handle days with extreme inflows, e.g., due to rain events. Another

advanced option is to simulate WWTP closure or upgrades by providing an end-year of operation for a WWTP
instance.
The main flows within the treatment section are influent, evaporation, and effluent, as described below.

Influent inflows
According to the basic model setup, excess wastewater beyond the plant's daily treatment capacity is discharged
to the predefined outflow location (see Table 4). However, the model holds advanced modeling capabilities,
enabling WWTP to accept larger inflows to handle temporal fluctuations (e.g., due to significant rain events).
Inflows higher than the designed capacity shorten the hydrological retention time (HRT or residence time),
resulting in less effective wastewater treatment. The designed retention time is calculated as $HRT_j^{Design} =$
$Volume_j/Inflow_j^{Design}$, where $Volume_j$ is the volume of WWTP $j$, and $Inflow_j^{Design}$ is the daily treatment capacity
of WWTP $j$ (Pescod 1992). The daily treatment capacity and time (or designed HRT) are model inputs (see Table
4). The minimally allowed HRT (days) parameter allows treatment plants to maintain higher inflows than their
designed capacities. It expresses the lowest operational hydraulic retention time a treatment plant can withstand
before it refuses inflows. Following the calculation of the hydraulic retention time, the maximum daily capacity
can be calculated as follows $Inflow_{max} = Volume/HRT_{min}$, whereas volume is fixed. For example, a minimally
allowed HRT of 0.8 days implies an increase of 25% in the operational daily capacity for a designed treatment
time of 1 day.

Evaporation
Water surface evaporation is calculated by multiplying the potential open water evaporation rate with the treatment
pools' estimated surface area, and the pool live storage volume limits it. Calculating the surface area of the
treatment pools is different for intensive and extensive systems. The surface area of an intensive WWTP is defined
as the ratio between the plant volume and the pool depth. For that purpose, a simplified representation of a WWTP
treatment pool is adopted based on a clarifier design (used during both primary and secondary treatment; Pescod,
1992), and the pool depth is estimated at 6 meters (WEF, 2005; see Figure B1).
Extensive systems are modeled as natural biological treatment ponds, alternately filling up and treating water.
These processes consist of a relatively short anaerobic treatment in deeper ponds followed by a long-term (20-40
days) residence in facultative shallow ponds (see Figure 1B; also refer to Pescod, 1992). Unlike intensive systems,
treatment ponds in extensive systems may remain empty for long periods. Since evaporation is simulated at the
pond level, it considers only ponds with positive water storage.

**Equation 2: calculation of the surface area of extensive treatment systems.**

$$As_j = \frac{1}{Depth_j} \times \left( VolCap_j \times \frac{TreatTime_j}{TreatPool_j - 1} \right)$$


The surface area of each treatment pool is calculated by dividing the pool's volume by its depth (see Equation 2;
*Depth*, currently set to 1.5 meters, as the depth of a facultative pond; Pescod, 1992). Each pool volume is derived

by multiplying the daily capacity (*VolCap*) with the pool filling time. The latter is a function of the designed treatment time (*TreatTime*) and a predefined number of treatment pools (*TreatPool*; currently set to two; Pescod, 1992). Although evaporation losses are overall small (see Figure 4), we allow modelers to change these default technical values with their estimates (see Appendix B).

## Effluents

Treated wastewater (effluents) are discharged into a natural water body or sent to reservoirs for reuse. The timing of effluent release differs between intensive and extensive systems. Figure 1B shows the main differences between these two types of systems. In intensive systems, influents remain in the treatment plant throughout the predefined treatment time. For example, for a treatment time of one timestep, the effluent volume at time $t$ equals the influent volume minus evaporation of time $t – 1$.

Extensive systems differentiate between two types of treatment ponds. At each time, one treatment pond receives all inflows; the other pond is either full or empty. Ponds that do not receive inflows and are not empty are considered 'active', i.e., in which wastewater treatment occurs. Effluents are released from 'active' ponds under any of the following conditions: (a) the predefined treatment time has passed since the 'active' pond stopped receiving inflows; (b) all pools are at full capacity, and more influents should be added into the system. In the latter case, the effluents always originated from the 'active' pond that had gone through the longest treatment time, though they may not be fully treated.

### 2.2.3. Post-treatment

The basic module has two post-treatment options: river discharge and reuse. Direct reuse (e.g., for irrigation, industrial, and potable uses) is possible using the CWatM reservoirs and water demand routines. This option requires data on the linkages between WWTP and reservoirs, representing existing or planned water conveyance systems. The routine iterates over the list of WWTP-Reservoir links and attempts to send treated wastewater to associated reservoirs. In the case of multiple receiving reservoirs, the water is split in proportion to the reservoirs' remaining storage (calculated as $remainingStorage_{i,t} = totalVolume_i - liveStorage_{i,t}$). Access water is discharged on predefined overflow locations if all related reservoirs are full. Discharge into streams/rivers is the default behavior if no reservoir is associated with a treatment plant. Finally, untreated wastewater is discharged if a plant's inflows exceed the plant's peak capacity (see minimally allowed HRT in section 2.2.2).

Treated wastewater can be managed in a separate reuse system by establishing a set of artificial, off-stream (type-4) storage reservoirs. A type-4 reservoir is not connected to the river network, thus having no channel-related inflows or outflows. Instead, water inputs include water/wastewater pumping, and water outputs are evaporation and pumping. The model combines the two approaches mentioned above, as each WWTP can be linked to one or more reservoirs or discharge its water directly into a river channel. Indirect reuse can be simulated by releasing the water into a channel, upstream to a lift area where river water is abstracted and used, or into a reservoir linked to the river network, where effluents are mixed with fresh water.

The module is designed to allow inter-basin transfers of wastewater or treated wastewater, yet this advanced option is not required in the case of a global model. Interbasin transfer of treated wastewater aims to account for cases in which the reuse areas extend beyond the borders of the simulated river basin. In that case, WWTP-specific export-share parameters indicate the daily fixed percent of treated wastewater transferred for reuse in other basins.

Similarly, the interbasin transfer of un-treated wastewater represents cases in which treated wastewater collected
in one basin is treated in another. It occurs automatically if a defined service area is not associated with any
WWTP within the simulated basin.
## 3. Case study application
Israel is located on the Eastern Coast of the Mediterranean between the latitudes 29°N –34°N and along the 35°E
longitude. Its Central coastal and Northern parts are governed by a Mediterranean climate (hot and dry summer),
its Eastern parts are arid due to rain shadow from its Central Mountain range, and the Southern parts experience
a semi- to hyper-arid climate due to their vicinity to the world's desert belt.
During the 1960s, Israel initiated a country-wide water conveyance system (the 'National Water Carrier') to
transfer water southwards from the northern Sea of Galilee, allowing rural development and large-scale irrigation
in the semi-arid Negev region (Tal, 2006). Israel's water system is intensively managed today and relies primarily
on seawater desalination, treated wastewater reuse, and groundwater abstraction. Although it is a nationally
managed system, significant regional differences exist in sectoral water provision (Fridman et al., 2021).
The Ayalon basin is in central Israel and the West Bank and stretches 815 km$^2$ between the western slopes of the
Judea Mountains and the Mediterranean Coastal zone. A few kilometers inland, the Ayalon spills into the Yarkon
stream (see Figure 2). Ayalon is an urbanized river basin partially overlaying the Tel Aviv-Yaffo metropolitan
area downstream and the city of Modi'in in its middle segment. Downstream urban areas result in considerable
water demand, vast runoff from sealed areas, and a high rate of wastewater generation. Upstream, the landscape
of the Ayalon basin is predominantly a rural mosaic of open areas and small settlements. Patches of irrigated
agriculture and forests are primarily found in the South-Eastern parts of the basin.
Ayalon is a seasonal river originating in the South-Eastern part of the basin. An artificial 'horseshoe' shaped
reservoir ('Mishmar Ayalon') regulates its flows and maintains relatively fast groundwater recharge. Five main
tributaries drain the remaining basin and feed the Ayalon River downstream. An artificial cemented canal collects
the river water before crossing densely populated urban areas downstream.
### 3.1. Data sources
The CWatM provides global datasets at 0.5 degrees and 5 arc-minutes, as Burek et al. (2020) described. This high-
resolution analysis better combines global and local data sources to represent the case-study hydrologic processes
and human-hydrologic interactions (Hanasaki et al., 2022). Table 1 provides an overview of both the global (e.g.,
meteorological forcings, soil characteristics, topography, and the river network) and the local datasets (e.g.,
wastewater treatment and reuse, reservoir networks, aquifer properties, landcover maps, seawater desalination,
and water demand). A complete documentation of the dataset associated with this publication is available at
https://doi.org/10.5281/zenodo.12752967.

**Table 1: Model inputs from global and local datasets. Unless explicitly indicated, all datasets were resampled to 30 arc-**
**seconds or converted to a raster format.**

| Input data | Spatial resolution | Temporal resolution | Data sources and comments on data processing |
|---|---|---|---|
| | | | |

| Global datasets | | | |
|---|---|---|---|
| Meteorological forcings | 0.5° grid | Daily | ISIMIP 3a, GSWP3-W5E5 (Lange, Mengel, Treu, & Büchner, 2022) |
| Spatio-temporal precipitation and temperature patterns for downscaling | 30 arc-seconds grid | Multi-annual monthly average | WorldClim (Fick & Hijmans, 2017) |
| Soil | 30 arc-seconds grid | Fixed value | Dai et al. (2019) Shangguan, Hengl, Jesus, Yuan, & Dai (2017) |
| Topography | 3 arc-seconds grid | Fixed value | MERIT Digital Elevation Model (Yamazaki et al., 2017) |
| River network properties flow direction map | 30 arc-seconds grid | Fixed value | MERIT Hydro IHU (Eilander et al., 2020) |
| Local/modified datasets | | | |
| Landcover maps | 500 meters grid | Annual | MODIS Global landcover between 2001 -2019 (Friedl & Sulla-Menashe, 2019), OpenStreetMap (Urba areas, water, and green spaces; available at https://www.openstreetmap.org), Ministry of Agriculture and Rural Development (MOAG, 2022; cultivated land), and Hamaarag (2017; forests' map) |
| Municipal and industrial water demand | Local government borders, polygons | Annual | Israel Central Bureau of Statistics (ICBS, 2022). A Random Forest regression imputed missing data for different localities and specific years. Palestinian Central Bureau of Statistics (PCBS, 2022). |
| Wastewater treatment plant location | Point data | Fixed value | A national dataset was compiled mainly relying on a report by the Israel National Reserve Authority (INRA, 2016) and data from PCBS (2022a). Wastewater treatment plants' discharge points (e.g., due to overflow) are fixed to the WWTP location. |

| | | | |
|---|---|---|---|
| Wastewater attributes and technical data | Tabular format | Annual | A national dataset was compiled mainly relying on a report by the Israel National Reserve Authority (INRA, 2016) and data from PCBS (2022a).<br><br>Attributes include wastewater treatment levels', and years of operation. |
| Wastewater collection systems | Local government borders, polygons | Fixed value | A national dataset was compiled mainly relying on a report by the Israel National Reserve Authority (INRA, 2016) and data from PCBS (2022a).<br><br>The data for the wastewater collection systems include service areas, connection rate, and wastewater generation coefficients. |
| Desalination | National value | Annual | Annual desalination capacity between 2005 -2019 (Gov.il, N.D.). A basin-scale desalination is allocated proportionally to the relative domestic water demand. For example, the national supply of desalinated seawater in 2005 and 2015 was 20 and 503.4 MCM, respectively. In the same years, the Ayalon desalinated seawater supply is estimated at 3.4 and 88 MCM. |
| Reservoirs | Digitized polygons and attributes | Fixed value | Manually identify and digitize reservoirs based on aerial photography and satellite imagery. Depth and volume were assumed based on fieldwork and engagement with water managers. The link between WWTP to reservoirs is based on INRA, (2016). |
| Aquifers delineation | Digitized polygons | Fixed value | Israel Hydrological Services (2014), |
| Aquifer properties – coastal aquifer | Digitized polygons | Fixed value | Melloul et al. (2006).<br>Aquifer properties include porosity and permeability. |
| Aquifer properties – mountain aquifer | Digitized polygons | Fixed value | Wollmann, Calvo, & Burg (2009). |

| | | | Aquifer properties include porosity and permeability. |
|---|---|---|---|


## Groundwater basins and aquifers

This case study uses the coupled CWatM-MODFLOW6 model to account for the interface between surface and
groundwater hydrology and groundwater dynamics (Guillaumot et al., 2022). The Ayalon River basin lies above
two principal groundwater aquifers. The west mountain aquifer is part of the larger Yarkon-Taninim aquifer
system and has two partially separated sub-aquifers reaching a thickness of 600 meters. It comprises carbonate
sedimentary rocks and has a relatively high but non-homogenous hydraulic conductivity (Wollmann, Calvo, &
Burg, 2009). The slopes of the West Judea mountains function as recharge zones, and the top layers in the Western
foothills are made of chalk and marl and act as an aquitard, confining the Western Mountain aquifer (see Figure
A1). To the west, the relatively shallow Coastal aquifer (thickness up to 200 meters) mixes a sandstone aquifer
with a clay lens, resulting in varying hydraulic conductivity (Melloul, Albert, & Collin, 2006). Data on
groundwater abstraction volumes, locations, and the water table changes was unavailable.

## Reservoirs

We have manually identified and digitized reservoirs in the Ayalon basin using multiple data sources, including
georeferenced aerial photography, visual inspection of satellite imagery, fieldwork, and interviews with local
water management experts. The biggest reservoir in the Ayalon basin is Mishmar Ayalon (7.5 MCM; Figure 2),
a seasonal water storage fed by the upstream section of the Ayalon River and regulates downstream flows. The
Natuf reservoir is located on a prior quarry site northeast of the basin (4.3 MCM) and contributes to groundwater
recharge. Four smaller reservoirs constitute the wastewater irrigation infrastructure and have a total designed
storage of 634,200 $m^3$. This reuse system extends beyond the basin's borders, for which we account by exporting
a fraction of the treated wastewater.

## Wastewater in the Ayalon basin

Two primary wastewater treatment plants collect wastewater generated in the main cities, and small-scale
treatment plants collect those generated in the rural sector. The Shafdan WWTP treats all wastewater generated
in the Tel Aviv-Jaffa metropolitan area in the adjacent Sorek basin, which is out of the scope of this analysis.
Later, they were exported to the North-Western Negev for irrigation purposes (Fridman et al., 2021). The Ayalon
WWTP is the most significant facility in the basin, with a daily capacity of 81,000 $m^3$. It collects treated
wastewater from the cities of Lod and Modi'in (see Figure 2) and their surroundings. An extensive treatment plant
has existed since 1995, but development and population growth have exceeded its capacity, increasing sewer
discharge frequency into the stream. An intensive activated sludge treatment plant with a daily capacity of 54,000
$m^3$ started operating in 2003. However, on some occasions, the daily inflow exceeded the daily capacity by over
1.5 times (see Table S4). Almost ten small-scale wastewater treatment plants in the Ayalon basin are treating
sewers at a settlement scale with a total daily capacity of 12,298 $m^3$.


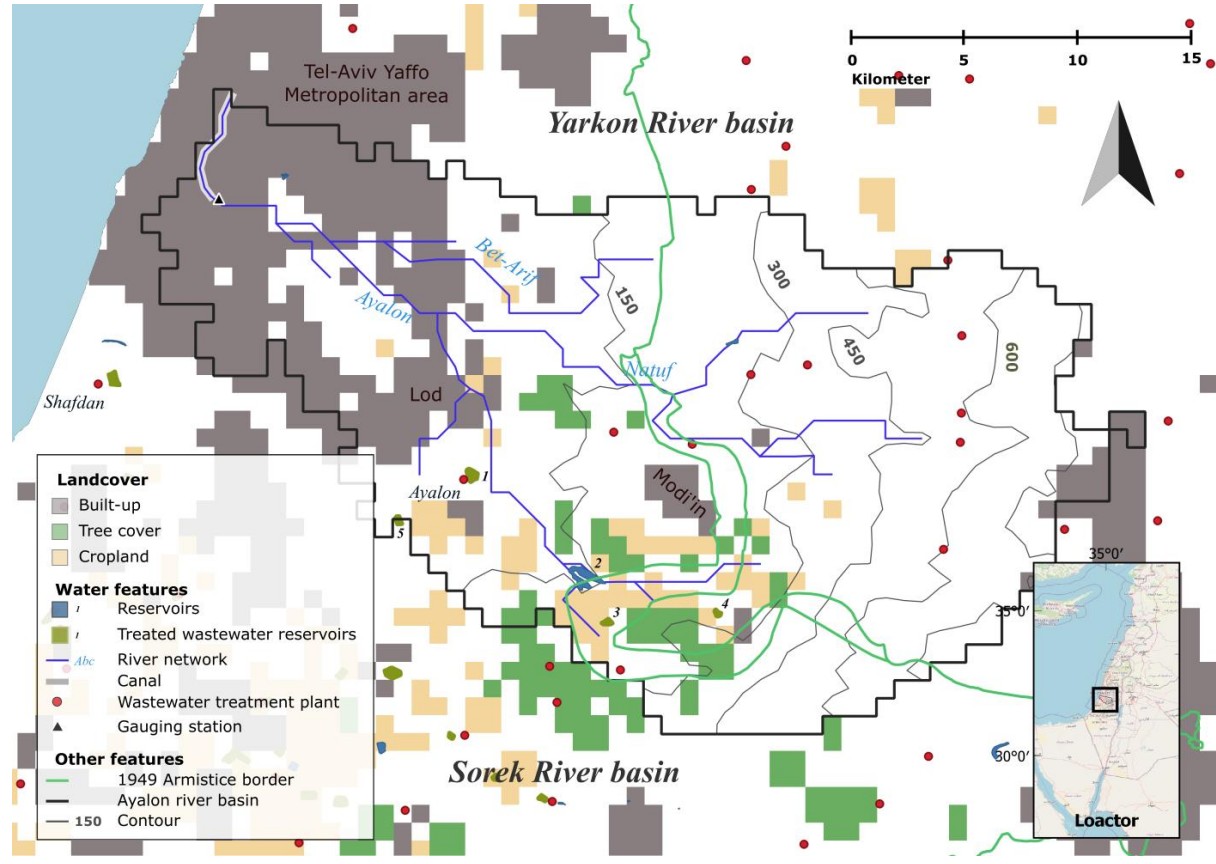


**Figure 2: the Ayalon River basin case study: land cover and significant water features. Partially uses data from ©**
**OpenStreetMap contributors 2022. Distributed under the Open Data Commons Open Database License (ODbL) v1.0.**
**Marked reservoirs: (1) Ayalon; (2) Mishmar Ayalon; (3) Ta'oz; (4) Mesilat Zion; (5) Mazli'akh.**

### 3.2. Setting calibration scenarios and model parameters

In this analysis, we simulate the Ayalon basin hydrology and wastewater treatment and reuse under three scenarios, aiming to explore the effects of the wastewater treatment module's different modes of operation on model calibration and basin-scale water resource management. In the first scenario (S0), we disable the wastewater treatment and reuse module. The second (S1) and third (S2) include wastewater treatment and reuse without and with urban runoff collection, respectively. The share of urban runoff flowing into the sewers is set as a calibration parameter in S2. In this case study, we defined sectoral water allocations to limit wastewater reuse to irrigation, with limited use for livestock purposes. Additional calibration parameters are associated with evapotranspiration rates of irrigated croplands and grassland, soil depth adjustment, within grid-cell soil moisture spatial distribution, soil hydraulic conductivity and water content at saturation, Manning's roughness coefficient, riverbed exchange rate, urban evaporation coefficient, and urban infiltration coefficient. The emphasis on the urban landscape is due to the relatively high share of built-up areas in the Ayalon basin (see Figure 2).

We set three more wastewater reuse scenarios apart from calibration scenarios by expanding the irrigated agriculture area (by 2.5%) and increasing storage volume (by 5%) for two reservoirs for which command areas are defined: Ayalon and Mazlikh. One scenario includes expansion and increased storage, and each of the other scenarios includes expansion or increased storage.

## 4. Results

### 4.1. Model validation

We have calibrated the Ayalon case study against the daily average discharge at the Ayalon-Ezra gauging station (34.794° E, 32.04° N; Figure 2) over the period August 1st, 2001, to July 30th, 2006, and validated over the period August 1st, 2007 to December 31st, 2019. We further compared the simulated evapotranspiration with multiple satellite-derived products (Figure S7; Mu et al., 2014; Reichle et al., 2022; Rodell et al., 2004) and the simulated monthly influent flows into the Ayalon WWTP with observed data between 2016 -2019 (Figure 5; Ayalon Cities Association, 2021). We measure model performance using the Kling-Guphta Efficiency (KGE) and Nush-Sutcliffe equilibria (NSE) coefficients (Moriasi et al., 2015).

The S2 (wastewater and urban runoff collection) scenario generated the best-performing model (KGE = 0.76; NSE = 0.72 during training), followed by S1 (wastewater without urban runoff collection; KGE = 0.27; NSE = 0.61), and S0 (KGE = -0.4; NSE = 0.57). Model performance is lower during the validation periods across all scenarios. During the validation periods, the complete implementation in scenario S2 also resulted in the best-performing model ($KGE_{2006-2013}$ = 0.69, $KGE_{2014-2019}$ = 0.55). Over the complete simulation period (1995 -2019), the mean observed discharge at the outlet is 0.81 $m^3$ $s^{-1}$, and it was best matched by the simulated discharge in scenario S2 (0.87 $m^3$ $s^{-1}$; see Table 3). The full implementation scenario (S2) best matches the observed discharge during most days in the dry (April-September) and the wet season, as demonstrated in Figure 3B. Sometimes, the model overestimates discharge or simulates flow events during the dry period (e.g., late April 2003, see Figure 3C). This overestimation is often associated with a mismatch between forcing data (e.g., precipitation) and actual precipitation (see Figure S6 and Table S1). The S2 scenario performs well and captures peak events better when compared to the alternative modes of operation. For example, it overestimated the discharge in a peak flow event at the end of February 2003, whereas others underestimated the discharge by over 50% (see Figure 3A and B).

The simulations were compared with different remote-sensing derived evapotranspiration (RS-ET) time-series. All scenarios can capture seasonal dynamics but overestimate ET during early spring (around March-April, except SMAP; see Figure S7). The 'No wastewater' (S0) scenario highly overestimates the ET, whereas the other two (S1, S2) scenarios better align with the RS-ET data, particularly after 2015. There are differences between RS-ET datasets associated with process, forcings, and parameterization errors (Zhang et al., 2016); some are shown in Table S2. GLDAS v2.1 shows the lowest KGE across scenarios, and SMAP indicates the highest (see Table S3). These findings are consistent with an intercomparison of RS-ET datasets (Kim et al., 2023). Furthermore, the fitness to RS-ET time-series improves when additional features of the wastewater module are incorporated across all datasets. The average KGE is -0.68 (S0), -0.27 (S1), and -0.17 (S2).

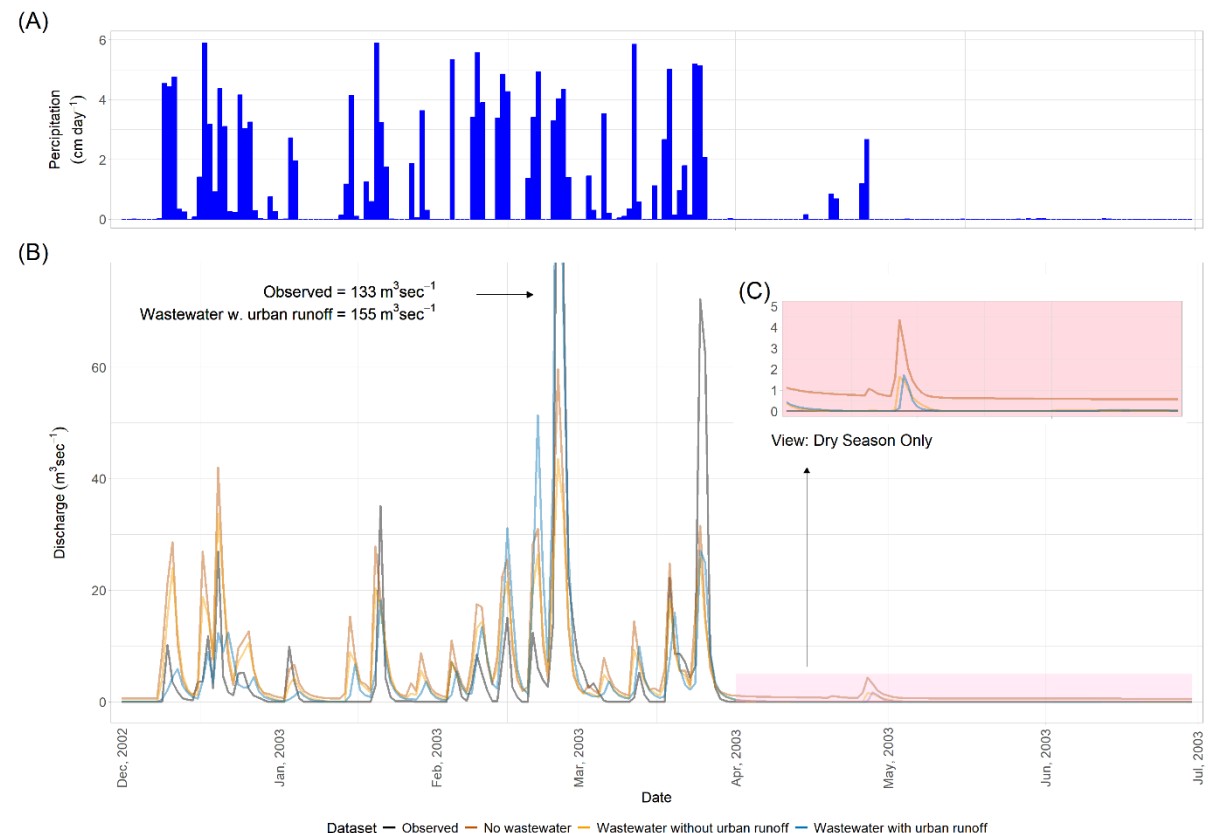

338

**Figure 3: (A) Daily average rain depth in the Ayalon River Basin, and (B) observed and simulated discharge at the outlet between December 2002 and July 2003. (C) Zoom in to the observed and simulated discharge in the dry season.**

Modeling the intermittent Ayalon River case study is challenging, mainly due to its arid climate and small basin area. Under these conditions, even a small deviation in the absolute simulated discharge results in a high relative error. It follows that diverting return flows (i.e., sewage) away from the river was a crucial step in the Ayalon model calibration. Introducing wastewater treatment and reuse into CWatM enables simulating actual water dynamics in the Ayalon basin, resulting in a better-performing model. The KGE values of scenarios S0-S2 between 1995 and 2019 are -0.75, 0.17, and 0.66, and the percentage differences between the simulated and observed average discharge are 162%, 62%, and 4.1%,  respectively (see Table 3). Similar improvement is also shown when comparing simulated and observed discharge between 1995 and 2019 (see Figures S1, S2, and S3). The improvement from including the wastewater treatment and reuse module (scenario S1) is associated with reducing the dry season's baseflow from an average of 0.07 $m^3$ $s^{-1}$ to 0.06 $m^3$ $s^{-1}$. The effects of urban runoff collection were mainly evident in the wet season's discharge, which was reduced from an average of 2.53 $m^3$ $s^{-1}$ (scenario S1) to 1.68 $m^3$ $s^{-1}$ (scenario S2). The collection of urban runoff into the sewers reduces flows downstream to urban areas and fits, to some extent, the inflow dynamics into the Ayalon wastewater treatment plant (see Figure 5).

**Table 3: Model performance under different scenarios over the complete simulation (1995 -2019). The dry season occurs from April to September.**

| Scenario | KGE | NSE | Annual mean discharge | Dry season's mean discharge | Wet season's mean discharge |
|---|---|---|---|---|---|
|  |  |  |  |  |  |

|  | (during calibration) | (during calibration) | (% relative to observed) | (% relative to observed) | (% relative to observed) |
|---|---|---|---|---|---|
| Observed | - | - | 0.81 ± 4.9 (-) | 0.04 ± 0.38 (-) | 1.59 ± 6.9 (-) |
| S0: No wastewater | -0.75 (-0.4) | 0.55 (0.55) | 2.12 ± 5.1 (162%) | 0.7 ± 0.85 (1650%) | 3.54 ± 6.92 (123%) |
| S1: Wastewater without urban runoff collection | 0.17 (0.27) | 0.62 (0.61) | 1.3 ± 4.36 (62%) | 0.09 ± 0.65 (125%) | 2.53 ± 5.9 (59%) |
| S2: Wastewater with urban runoff collection | 0.66 (0.76) | 0.7 (0.72) | 0.87 ± 4.1 (7%) | 0.06 ± 0.42 (50%) | 1.68 ± 5.67 (6%) |

357

## 4.2. Component and flows of the wastewater module

The wastewater flows between different model components are illustrated in Figure 4 using the water circle concept. A water circle is a simplified depiction of the water cycle within a specific region, component, and timeframe. It illustrates the water balance by linking inputs, outputs, and changes in storage while representing various water sources and uses (Smilovic et al., 2024). Figure 4 presents the wastewater reuse water balance in the Ayalon River basin between 2001-2006, totaling 209 million cubic meters per year (Inputs + Outputs + Change in Storage). Inflows to wastewater treatment plants primarily originated from non-irrigation return flows (labelled as 1 in Figure 4), consisting mainly of domestic sewage mixed with urban runoff, especially in dual-purpose urban drainage systems. These inflows are based on existing model routines (e.g., water demand and soil; see Figure 1A) and amount to 104 MCM. In the Ayalon basin case study, the largest share (almost 70%) of the influents is being treated in the Shafdan WWTP outside of the basin of interest (labelled as 2 in Figure 4 and Figure 2), and approximately 14% are sent to reservoirs for reuse, though actual reuse is lower (labelled as 4 in Figure 4). The gap between the volume of wastewater sent to reservoirs and the actual reuse is associated with evaporation, outflows, and leakage losses (prominent in one of the reservoirs, see Figure S4). The remaining share includes the discharge of treated wastewater (4%) and raw sewage (8%; labelled as 4 in Figure 4). Evaporation loss from WWTP is marginal (<4%) and is represented by one of the unlabeled wedges on the wastewater circle (labelled as 5 in Figure 4).

The annual average wastewater reuse in the Ayalon basin (2.3 MCM) accounts for almost 10% of the basins' irrigation withdrawal (25 MCM). In addition, around 71 MCM of the wastewater generated in the Tel-Aviv metropolitan area (see Figure S10), are treated in the Shafdan WWTP (in the Sorek River Basin) and reused for irrigation in the South of Israel (Fridman et al., 2021).

379

380

381

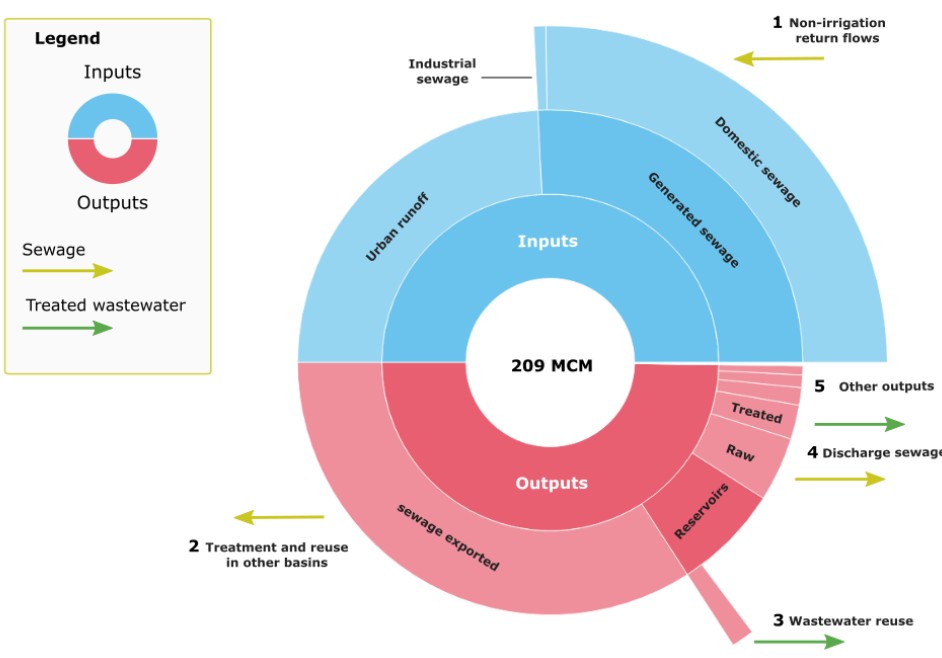

382

**Figure 4: Average annual sewage and treated wastewater flows within and between CWatM model components (see labels 1-5), based on a simulation for the Ayalon River Basin, Israel, from 1/1/2001 -30/07/2006.**

### 4.3. Modeling wastewater and urban stormwater collection systems

CWatM includes two main hydrological processes for urban areas: return flows (e.g., sewage generation) and urban runoff. These flows are managed by either separated or combined collection and drainage systems. In Israel, two systems are operated separately to collect urban wastewater and stormwater. However, stormwater frequently leaks into the sewers due to illegal connections of urban drainage.

The runoff collection coefficient allows the user to control the magnitude of systems integration. One combined system would have a coefficient of one, implying all urban runoff flows into the sewers collection system, and a coefficient of zero suggests two completely separated systems. The calibrated model ended up with a coefficient of 0.78, implying that 78% of urban runoff flows into the sewers.

The advantages of the runoff collection coefficient are shown in Figure 5, comparing the monthly inflows to the Ayalon WWTP against the simulated inflows with (S2) and without (S1) urban runoff collection. On average, between 2016 and 2019, the Ayalon WWTP accepted 1,780 +/- 86 thousand m$^3$ sewers every month. The average inflows in the scenarios without and with urban runoff collection are 1,562 +/- 119 and 1,699 +/- 203 thousand m$^3$ per month, respectively. Overall, the model underestimates the inflow to the Ayalon WWTP, as shown in the top panel of Figure 5, during the dry months (e.g., April to June), which is probably due to the use of annual model inputs for water withdrawal, that do not capture seasonality. Seasonality is only captured by the 'Wastewater with urban runoff' (S2) scenario as a direct result of urban runoff collection. Another factor limiting WWTP inflows is the minimally allowed HRT presented in section 2.2.2. Sensitivity analysis implies that a one percent change in the parameter value results in an average 0.23% change in the WWTP inflows (see Supplementary Information and Figure S9).

Rain events during the wet season often result in increased inflows into the wastewater treatment plants (e.g.,
during December 2016 or January 2018). The scenario that includes urban runoff collection (S2) can simulate
these peaks, though it slightly overestimates them, whereas no peaks are simulated for scenario S1, where no
urban runoff is collected (see Figure 5 bottom panel). While it may be that the runoff collection parameter was
set at a value that is too high, overestimating the peak flows can also result from errors in precipitation data (see
Figure S6). The wastewater with urban runoff collection (S2) scenario performs the scenario without wastewater
collection based on multiple parameters (showing lower bias and higher NSE and correlation; see Table S5).

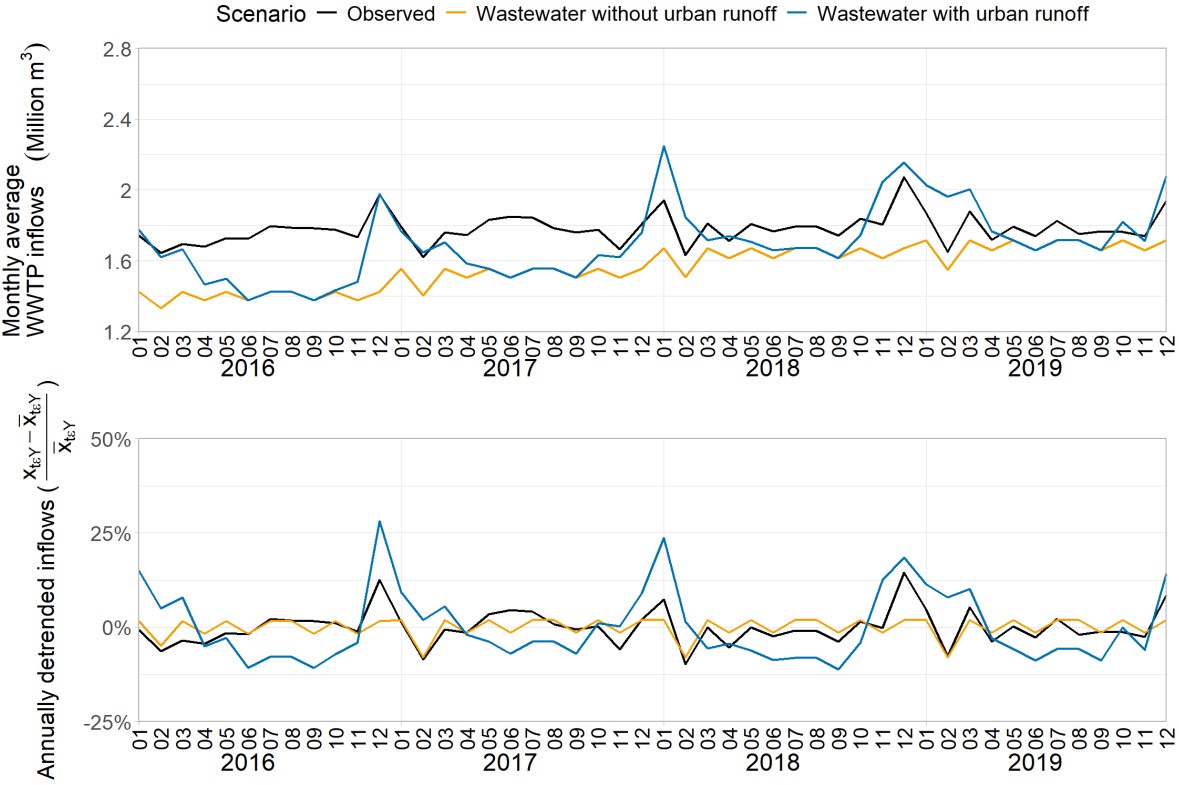


**413 Figure 5: Observed VS. simulated monthly wastewater inflows into the Ayalon WWTP with and without urban runoff**
**414 collection using absolute values (top chart) and annually detrended values (bottom chart).**

415          **4.4. Modeling of wastewater reuse potential and impacts**

Wastewater treatment and reuse may significantly affect water management, particularly for complex water
resource systems in water-scarce countries. Israel is a water-scarce country that reuses wastewater, utilizes
desalination water, and transfers water between river basins to mitigate water stress. As Israel manages water
nationally, analyzing water resources on a basin scale aligns differently from Israel's actual state of water
resources. Instead, the following scenarios aim to illustrate the relevance of the WTRM module to water resource
management.
Until the early 2000s, the Ayalon River basin's water supply relied primarily on groundwater abstraction. As a
result of population growth and the expansion of the Ayalon WWTP's daily treatment capacity in 2003 (from
22,000 to 54,000 m3/day), the simulated wastewater **reuse** has nearly doubled, increasing from 1.5 million $m^3$ in
the year 2000 to 2.7 million $m^3$ in 2005. In the same year, desalinated seawater was first supplied, satisfying
approximately 3% of the total water demand in the basin. Over the years, the role of desalination increased,

accounting for around 47% of the water supply. The share of treated wastewater slightly increased, reaching 2.7% (approximately 3 million $m^3$), compared with 1.5% in 2000. Most importantly, avoided groundwater pumping in 2010 enhanced Israel's water security by reducing the pressures on aquifers, and the avoided seawater desalination reduced energy-for-water use and water production costs (Fridman et al., 2021).

Focusing on irrigation districts linked to the Ayalon WWTP(see Figure S11), Table 3 presents the multiannual average absolute and relative wastewater **reuse** (for irrigation) between 2000 -2010. Overall, there is little difference between the baseline and agricultural expansion scenarios, showing a slight increase in the **reuse** volume but a slight decrease in the relative wastewater irrigation (relative to irrigation demand). These findings point out a balanced proportion between storage and water demand. Small access storage is kept, allowing additional irrigation to respond to increased water requirements. The two scenarios, which include increasing storage, demonstrate higher wastewater **reuse** volume (4.7%-4.9%) and relative irrigation increasing from 17.3% to 17.8-18.1%. The share of wastewater **reuse** out of the total irrigation demand increases from around 13% to 18% in 2000 and 2003, respectively, and reached almost 25% in 2006 (see Figure S12). These changes were associated with an increased capacity of the Ayalon WWTP in 2003 and precipitation variability, e.g., lower irrigation requirements during wet years compared with a relatively constant supply of treated wastewater. As this **reuse** project extends southwards, outside the Ayalon River basin, the model also estimates additional wastewater **reuse** of almost 2 million $m^3$ (i.e., treated wastewater sent for **reuse** outside the basin). In addition, more than 50 million $m^3$ are collected and treated in the Shafdan WWTP southwest of the Ayalon river basin (see Figure 2) and are almost entirely reused.

**Table 3: Average and standard deviation of the absolute and relative wastewater reuse in irrigation districts linked to the Ayalon WWTP between 2000-2010.**

| Scenario | Wastewater reuse, thousands $m^3$ (share increase relative to baseline) | Wastewater irrigation (% of total irrigation) |
|---|---|---|
| Wastewater and urban runoff collection (Baseline) | 2,423.4±536.8 (-) | 17.3±4.1% |
| Agricultural expansion and increased reservoir capacity | 2,543.1±514.4 (4.9%) | 17.7±4% |
| Increased reservoir capacity | 2,536.7±515.3 (4.7%) | 18.1±3.9% |
| Agricultural expansion | 2,447.2±507.4 (1%) | 17±4% |

## 5. Discussion

**Wastewater treatment and reuse play a crucial role in the hydrological modeling of urban watersheds, especially in low-discharge/intermittent rivers.**

Discharges from wastewater treatment plants often dominate urban watersheds' hydrological signals, increasing low-flows, flashiness, and the frequency of medium and high-flow events (Coxon et al., 2024). The effect of

wastewater on stream hydrological signals would become more pronounced in intermittent streams, challenging
model calibration. Acknowledging this fact, one may compromise on model performance in urban watersheds,
yet including wastewater treatment and reuse in the modeling allows for increased model performance as it better
represents local water management processes. The example provided in this paper demonstrates this point by
showing a significant increase in model performance due to including wastewater treatment and reuse in the
modeling.
To our knowledge, only a few existing hydrological models account for wastewater treatment and reuse. Dyn-
Qual, for example, simplifies the treatment process and only allows for indirect reuse, i.e., treated water is
discharged into rivers and can be abstracted downstream. SWAT model represents wastewater treatment by
including pit latrines, yet both models focus on the water quality and missing critical operations associated with
water quantity (e.g., reuse through reservoirs or directly to fields). Although addressing the highly relevant topic
of water quality, the representation of wastewater processes in these two models would not contribute to model
calibration in urban or intermittent watersheds.
The importance of including wastewater treatment and reuse in high resolution (i.e., ~1km) hydrological modeling
is also aligned with recent findings, as these models are susceptible to the effects of human activity on the water
cycle and often require better representation of these processes and more precise data  (Hanasaki et al., 2022). It
follows that the WTRM complements the recent shift towards high-resolution modeling at global (van Jaarsveld
et al., 2024) and more local scales (e.g., CWatM implementation in Bureganland Austria; Bhima River Basin,
India; North China; Guillaumot et al., 2022; Yang et al., 2022).

**The wastewater treatment module utilizes multiple features of CWatM, providing tools to conduct policy-**
**relevant analysis on water resources management and wastewater treatment and reuse.**
Wastewater is increasingly perceived as an untapped resource and is marked as a potential water source to reduce
water stress or drought risk. Hydrological models, such as CWatM, are often used to inform decision-making and
policies for enhancing water resource management and can benefit from WTRM capabilities.
The WTRM interacts with different existing modules and routines in CWatM, allowing the modeling of different
wastewater reuse options. The 'source-sector abstraction fraction' and reservoir operation options in CWatM are
pivotal in modeling the treated wastewater reuse. The former is used to define the desired water mix, restricting
wastewater reuse by some sectors (e.g., forbidding households from using treated wastewater). Reservoirs allow
for the storage and transfer of treated wastewater and the reuse of it in relevant irrigation districts (i.e., by utilizing
the CWatM command areas feature). Leakage from reservoirs into groundwater (see Figure S4) can be used to
simulate groundwater recharge with treated wastewater.
Indirect reuse is enabled when treated wastewater is released into a river channel or a reservoir, diluted, and is
later abstracted downstream, and direct reuse is mediated through a designated reservoir, disconnected from the
river network (type-4 reservoirs). The inflows into this reservoir consist only of water transfers, and the outflows
are limited to abstraction and evaporation losses. The water levels in these reservoirs are not affected directly by
river flows and runoff, and they can maintain a traceable stock of treated wastewater over the long run. Abstraction
from reservoirs occurs either within a certain buffer (i.e., defined by the number of grid cells) from the reservoir
or within the area of an associated command area (area served by the reservoir regarding water supply). Combined
with the source-sector abstraction fraction, the modeling of the Ayalon basin has limited the use of treated
wastewater for irrigation and livestock to a smaller extent. Other existing uses, like urban landscaping or cooling
of thermal powerplants, were excluded, as data was unavailable.
By utilizing these modules and processes, the manuscript explores the potential effects of increased storage of
wastewater reuse reservoirs and expanding irrigated agriculture areas. It focuses on the command areas associated
with two reuse reservoirs (as indicated in Figure S11), indicating a high share of irrigation with treated wastewater
(~17%). The module variables could be utilized for exploring a wide variety of water management instruments,
including using treated wastewater to mitigate drought risk (conveying and storing treated wastewater in high
drought risk areas), to recharge the aquifer (controlling reservoir infiltration rate), or explore pathways for
agricultural expansion/intensification. Wastewater reuse can also have economic or environmental benefits. The
Ayalon case study is relevant for both due to potentially avoided seawater desalination, which is more expensive
and requires more energy. Considering the Nexus, economic, resource intensity, and emission data from different
sources (e.g., life cycle assessments; see Liao et al., 2020; Meron et al., 2020) could complement such analysis.

**Flexible model design and available global datasets provide a robust starting point for simulating**
**wastewater treatment and reuse scenarios at a global scale and coarser resolutions. Some data gaps remain**
**and provide opportunities for scientific engagement.**
The Community Water Model, as well as other large-scale hydrological models (Hanasaki et al., 2022; Hoch et
al., 2023), is shifting towards a multi-resolution modeling framework, allowing users to work on a global scale
with coarser resolutions and on a local scale with higher resolutions. The need to better represent wastewater
treatment and reuse in global, regional, and local hydrological modeling is linked to its increasing potential as a
water resource. The WTRM provides diverse tools for including wastewater treatment and reuse in hydrological
modeling. So far, the manuscript has focused on the module's advanced mode of operation, which is suitable for
data-abundant regions or local case studies where data collection efforts are feasible. Nevertheless, applying the
WTRM at coarser (e.g., 5 arc-minutes) spatial resolution globally or in data-scarce regions requires a simplified
workflow and a global data inventory.
Following the CWatM modular and flexible structure, the WTRM was developed with that notion in mind,
facilitating a simple mode of operation with minimal data requirements but including advanced processes when
data is available. The results presented and discussed show a significant increase in model performance as a result
of a more straightforward implementation of the module (i.e., without urban runoff collection), which, together
with the reuse scenarios, point to the potential impact of upscaling the analysis to cover other urbanized watersheds
and water-stressed regions. The recent development of different global datasets provides an opportunity for
upscaling this analysis, though these data would have to undergo some processing to fit the CWatM data structure.
Hydrowaste (Ehalt Macedo et al., 2022) is a global WWTP dataset describing plants' location, treatment level,
operational status, population served, overflow discharge point, and daily capacity. It was recently used to
determine the impact of droughts on water quality (Graham et al., 2024) and to account for the global microplastic
fiber pollution from laundry (Wang et al., 2024). Second, Jones et al. (2021) compiled a global gridded dataset (at
a 5 arc minutes resolution) describing wastewater generation volumes and collection, treatment, and reuse rates.
The data has already been used to force global studies on water quality (van Vliet et al., 2021).
These two datasets provide sufficient global data at a spatial resolution of 5 arc minutes to accommodate six of
the seven mandatory variables required to setup a simple simulation (see Table 4). However, data is lacking for
the year of establishment (or the start of operation) of a WWTP, which could be assumed by utilizing auxiliary
time-series data, like drinking water sanitation and hygiene (WASH) available from the joint monitor program
(JMP, at https://washdata.org), or sectoral outputs from monetary input-output tables (e.g.,
https://worldmrio.com). These data could cast temporal trends of increased sanitation coverage or sectoral
economic activity. Two additional challenges are indicated in Table 4, associated with the treatment days and
service (wastewater collection) area. In this study, we rely on a national dataset associating municipalities with
WWTPs (see Figure S10; INRA, 2016), yet this data is not available for most countries. Instead, following Ehalt
Macedo et al. (2022), the wastewater collection areas can be traced back from the WWTP to serve the nearest,
most likely upstream, population centers. Treatment days are associated with the WWTP classification into
intensive and extensive, which can be associated with location and economic factors (like GDP per capita or
electrification status). The availability of such data at national, sub-national, and grid scales deems the
classification of WWTP as intensive or extensive and feasible.
Advanced simulations are not pursued globally, so data sources for their required variables are not sought, except
for reuse and reservoir connections, as reuse significantly impacts model performance and water resource
management analysis. The reuse rates estimated by Jones et al. (2023) can be used for that purpose. However, as
it is not linked to any specific WWTP or reservoir, as required by the WTRM, it would require some pre-
processing and simplifying assumptions. Some ongoing efforts to identify potential wastewater reuse for specific
WWTP can support this processing (Fridman et al., 2023), yet both data sources would involve high uncertainties
at the grid scale. Two other approaches could be taken to assess different reuse scenarios, including indirect reuse
from waterbodies (e.g., rivers and lakes) or simulating on-site type-4 reservoirs with command areas set as fixed
buffers. Such reuse scenarios could be used to explore reuse by other non-agricultural sectors.
**Table 4: Model variables for simple and advanced simulations and potential data sources. Note: * indicates the variable**
**is unavailable but could be concluded by utilizing auxiliary data; ** indicates the variable is unavailable but could be**
**estimated based on published methods; *** indicates available data is highly uncertain at grid scale and can be used to**
**inform scenarios.**

| Model variable | Simulation mode | Description [Default value] | Potential Data source |
|---|---|---|---|
| Location | Simple | Geographic location (longitude, latitude) of WWTP [-] | Ehalt Macedo et al., 2022 |
| From year | Simple | The first year of a WWTP operation; as an advanced option, one may include the last year of operation (i.e., the closing of a treatment plant) or trigger several instances of a treatment plant (i.e., upgrade) [-] | Not available* |
| Volume | Simple | Daily capacity of the WWTP in cubic meters [-] | Ehalt Macedo et al., 2022 |
| Treatment days | Simple | Duration of treatment in days (retention time by design) is associated with treatment technology: intensive treatment (1 day) or extensive (approximately 30 days), as described in the manuscript [Intensive: 1 day; extensive > 1 day] | Ehalt Macedo et al., 2022* |

| | | | |
|---|---|---|---|
| Collection (service) area | Simple | Service area of different WWTPs, e.g., grid cells with water consumption which are connected to a given WWTP, indicated as WWTP ID [-] | Ehalt Macedo et al., 2022** |
| Collection share | Simple | Share of sewage generated, collected, and sent to WWTP, i.e., rate of connection (0 -1) to WWTP [-] | Jones et al., 2021 |
| Overflow | Simple | Geographic location (longitude, latitude) of the discharge point from WWTP into waterbodies (rivers, lakes, ocean) [-] | Ehalt Macedo et al., 2022 |
| Export share | Advanced | Share of treated wastewater used outside of the basin (0 -1; do not apply to global simulations) [0] | - |
| Contributing sectors | Advanced | Sectors from which wastewater is treated in a given WWTP (boolean 0/1) [1 for all sectors] | - |
| Min_HRT | Advanced | The minimally allowed hydrological retention time ranges between 0.001 -number of treatment days. This indicates how much additional water can be accepted daily over the daily capacity, e.g., in case of rain events or high water consumption. A value of 0.001 results in a potential inflow multiplier of 1,000, and a value equal to the treatment days results in no access inflows [treatment days] | - |
| Reuse and WWTP connection to reservoirs | Advanced | Links between WWTP and reservoirs and the rules for reuse of wastewater by different sectors [-] | Jones et al., 2021*** |


## 6. Conclusions

Wastewater primarily affects the hydrology in urbanized watersheds, particularly in water-stressed regions.
Wastewater reuse can ease the pressure on natural water sources and reduce drought risk. However, large-scale
hydrological models do not account for wastewater treatment and reuse. The recent trend towards higher spatial
resolutions further emphasizes the need to include local data and processes in hydrological modeling.
This paper introduces a novel wastewater treatment and reuse module integrated into the large-scale multi-
resolution Community Water Model. It provides a range of operational modes to balance modeling needs and data
availability worldwide. A high-resolution case study of an urbanized and water-stressed watershed illustrated the
WTRM's added value in terms of enhanced model performance and the inclusion of additional water sources,
such as reused wastewater. The role of wastewater in water resource management planning can now be included
in hydrological simulations, often used to inform such policies. Recently published global datasets were mapped
to model variables, indicating that global modeling at coarser spatial resolution (e.g., 5 arc minutes) is also
feasible. Some remaining data gaps, including the lack of time-series or missing information on reuse projects,
would require some assumptions and additional processing of input data. The compilation of a global input dataset
is one desired future development. As wastewater is naturally associated with water quality, this aspect remains a
limitation within the scope of the current development and would also be addressed in future developments.

## 7. Appendices

Appendix A

Figure A1 describes the vertical and lateral permeability of the YARTAN and coastal aquifers in Israel. The coastal aquifer forms a relatively narrow stripe stretching North to the South. Next, the western mountain aquifer is located towards the east, showing a relatively diverse permeability. The YARTAN groundwater basin includes the western mountain aquifer but extends far beyond the borders of the Ayalon River basin.

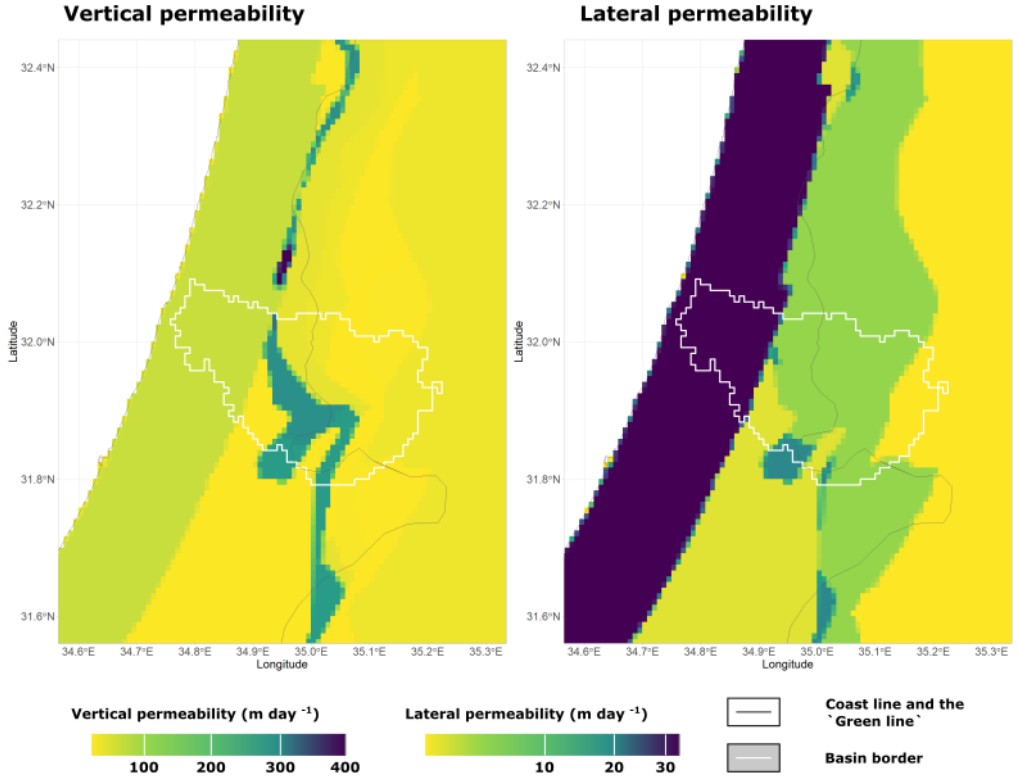

**Figure A1: Vertical and lateral permeability in the YARTAN and Coastal aquifers in the Ayalon basin and its surroundings.**

Appendix B

The treatment pool depth in an intensive WWTP represents the depth of a clarifier through which sewage flows at different treatment stages. The ratios between the clarifier's depth and diameter are relatively fixed to optimize sewers' biological treatment (e.g., bio-film development). A standard design for a clarifier is a relatively deep pool with a sloped bottom, as demonstrated in Figure B1. In the WTRM, the pool depth is only used to calculate the water surface area and simulate evaporation losses, and therefore, we find a simplified representation of the treatment pool with a flat bottom sufficient. In Figure B1, we convert the sloped bottom clarifier dimensions (WEF, 2005) to the equivalent pool depth in a flat clarifier, maintaining the pool's volume. This results in an approximate depth of 6.6 meters, which, based on data collected for the Ayalon case study, was rounded to 6 meters. We allow modelers to change the pool depth of either intensive, extensive, or both treatment systems by using the following settings in the settings file: 'pooldepth_intensive', 'pooldepth_extensive'. The default settings are hard coded as 6 and 1.5 meters, as described in this manuscript. In addition, to calculate the evaporation from

extensive WWTP, we allow users to change the default value of two treatment pools by adding the
'poolsExtensive' to the settings file.

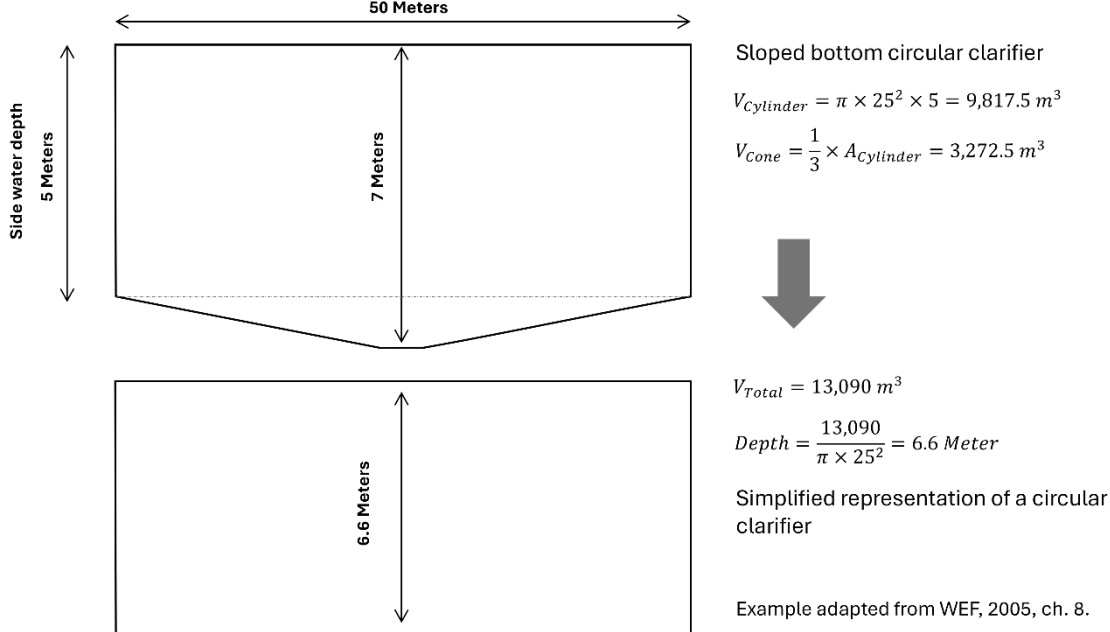

**Figure B1: A simplified approach to estimate wastewater treatment pool depth in an intensive WWTP.**

### 8. Code and Data Availability

The CWatM code is provided through a GitHub repository (https://github.com/iiasa/CWatM; last accessed:
February 15th, 2025), and the model version used for this study  (CWatM-Israel v1.06.1) is provided via
https://doi.org/10.5281/zenodo.13990296 (Fridman, 2024; last accessed: 25/10/2024). CWatM's documentation
and tutorials are available at https://cwatm.iiasa.ac.at/ (last accessed: February 15th, 2025). The input data used
for this publication, including model settings and initial conditions files, can be downloaded from
https://zenodo.org/doi/10.5281/zenodo.13990451 (Fridman et al., 2025; last accessed: 26/02/2025).

### 9. Author contributions

DF, MS, and PB developed the module code (Wastewater treatment and reuse) and prepared model input and
observation datasets. DF, MS, and PB have prepared model inputs. DF performed the simulations and associated
post-processing and prepared the paper with MS, PB, ST, and TK contributions. MS, ST, and TK acquired the
funding and undertook project administration and supervision.

### 10. Competing interests

The contact author has declared that none of the authors has competing interests.

## 11. Disclaimer

Any opinions, findings, and conclusions or recommendations expressed in this material do not necessarily reflect the views of the funding organizations.

## 12. Financial support

This research has received support from the SOS-Water project (Grant Agreement: 101059264) funded under the European Union's Horizon Europe Research and Innovation Programme and the GATWIP project funded by the Innovation and Bridging Grants Fund of IIASA. The work of Dor Fridman was partially funded by the IIASA-Israel Post-Doctoral program.

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
