# Peer review of "Wastewater matters: Incorporating wastewater treatment and reclamation into a"

_Geoscientific Model Development, 2024_

## Author Comment (AC4)

**Authors' responses to reviewer comments**

We thank the reviewers for their constructive feedback on our manuscript. Following are our responses to their comments. Blue color indicates our responses, *italic is used for citations from the manuscript*, and **bold shows implemented changes**. All the lines, figures, or tables mentioned below, refer to the revised manuscript.

The following paragraph summarizes our main changes, followed by a specific response for reviewers.

A major revision addresses the reviewers' comments and improve the manuscript. We are happy to submit the revised manuscript for your review and consideration. The revision aimed to improve the readability and clarity of the model description, further discuss the upscaling opportunities and module's applicability, improve methodological shortcomings, and enhance the analysis. Specifically, we have:

1. Revised the model development and description section and included a table describing the modules' variables.
2. We have emphasized the differentiation between the basic (or simple) and the advanced mode of operation. The case study demonstrated the latter, whereas the former fits a global analysis.
3. We have slightly modified the research design, forming a set of calibration scenarios (S0-S2, similar to those presented in the original manuscript) and reclamation scenarios to demonstrate the module's potential uses. We have recalibrated our model separately for each calibration scenario.
4. We revised the discussion section to focus on the upscaling potential, identifying currently available and suitable data sources, indicating data gaps, and proposing solutions to overcome them.
5. We have enhanced our analysis, particularly for the calibration and validation sections, and included some sensitivity analysis as supplementary material.
6. We add model settings for all the scenarios this manuscript presents as part of the supplementary material.

**Author responses to RC1**

The study introduces and describes a new "wastewater reclamation module" for the global hydrological model CWatM, evaluating the impact of the new module from a water quantity perspective by comparing observed and simulated river discharge for a case study location (Ayalon basin). This is an important area of research for (large-scale) hydrological models, with both aspects of water quantity and quality associated with wastewater have been largely overlooked and simplified in existing models. However, I find it somewhat difficult to fully follow the approach taken, and therefore I find that the manuscript would benefit greatly from a more comprehensive description of the three stages included in the workflow (i.e. pre-treatment; treatment; post-treatment), in addition to justifications for the assumptions made.

We greatly appreciate your constructive comments. We have clarified and simplified the module components and processes introduction, as presented in section 2.2 (lines 89-218). Specifically,

- We distinguish between two different modes of operation: a basic setup with a minimum data requirement, which is already feasible at the global scale, and an advanced (and optional) setup, which fits case studies where data availability is higher.
  In lines 89-95, we state:

*The wastewater treatment and reclamation module (**WTRM**) enhances the capacity of CWatM to simulate the human-water interface at **high** spatial resolution. It introduces wastewater generation, collection, treatment, discharge, storage, and reclamation to CWatM. **Large scale modeling shall utilize the basic setup of the WTRM for which sufficient data is available globally. Case studies for which data availability is higher, may benefit from a set of optional advanced function. The following section distinguishes between basic and advanced (optional) functionalities. Error! Reference source not found.**A demonstrates **WTRM** workflow**, split into three sub-processes**: (1) pre-treatment; (2) treatment; (3) post-treatment.*

- We describe all input data requirements (of the WTRM) associated with a simple/advanced simulation, potential data sources for global modeling, and default values (see Table 4, lines 555-559).

For example:

1. Line 115: "User-defined collection areas". What exactly does this mean/ how is the service area of a wastewater treatment plant defined? How applicable is the methodology for applying in areas lacking detailed information on wastewater treatment plants (i.e. export share, designed HRT)? While not tackled directly in this paper, the authors state that this work "sets the plans of "developing global input data for wastewater treatment and reclamation", but it seems to rely on plant-specific information that is not always readily available.

The term' *user-defined collection areas'* is indeed unclear. We have replaced it by the term' service area' and indicated that it is an input map (line 112).

***WWTP service areas (or collection areas) are model input that defines the linkages between location of wastewater generation (individual grid cells, denoted by l) to wastewater treatment plants (denoted by j).***

Some processes (e.g., minimum HRT) available via the WTRM are not intended to be used in a global setting. The distinction between the simple and advanced modes of operation is emphasized now in the 'Module development and description' chapter and particularly in section 2.2, Table 4 (see above), and in a revised section of the discussion where we state, for example (lines 519-524):

***Following the CWatM modular and flexible structure, the WTRM was developed with that notion in mind, facilitating a simple mode of operation with minimal data requirements, but including advanced procecsses when data is available. The results presented and discussed show a significant incerase in model performance as a result of a simpler implementation of the module (i.e., without urban runoff colleciton), which together with the reclmation scenarios, point on the potential impact of upscaling the analysis to cover other urbanized watersheds, and water stressed regions.***

The 'export share' is irrelevant for the global model, as it is only helpful if one needs to account for reclamation outside the simulated area (as noted in Table 4, lines 555-559).

Table 4, followed by a dedicated discussion of the model application, directly tackles the upscaling of the module and, to our taste, proves that global application is feasible. For example, in lines 524-534, we describe the following:

***Recent development of different global datasets provide an opportunity for upscaling this analysis, though, these data would have to undertake some processing to fit CWatM data structure. Hydrowaste (Ehalt Macedo et al., 2022) is a global WWTP dataset describign plants' location, treatment level, operational***

*status, population served, overflow discharge point, and daily capacity. It was recently used to deteremine the impact of droughts on water quality (Graham et al., 2024), and to account for the global microplastic fiber pollution from laundary (Wang et al., 2024). Second, Jones et al. (2021) compiled a global gridded dataset (at a 5 arc minuts resolution) describing wastewater generation volumes, and collection, treatement, and reclamation rates. The data has already been used to force global studies on water quality (van Vliet et al., 2021).*

*These two datasets provide sufficient global data at a spatial resolution of 5 arc minutes, to accommodate six out of the seven mandatory variables required to setup a simple simlation (see Error! Reference source not found.).*

2. How is seasonality in produced wastewater (and therefore reclamation) accounted for in the model? Line 127 suggests daily influent versus daily treatment capacity is considered; but does produced wastewater from e.g. domestic and industrial water uses actually vary at that temporal resolution? Similarly, in some areas, treated wastewater reuse may only occur intermittently throughout the year. How would the model deal with this/ is there a scheme for allowing temporal variability in wastewater reuse?

As CWatM simulates hydrological processes in a daily time step, the WTRM consistently operates at a similar resolution. Estimated wastewater generation (e.g., return flows) in CWatM (see lines 101-102 cited below) is a model input, usually available at annual or monthly (i.e., when considering the effects of temperature on water withdrawal) temporal resolution. The absolute amount of return flows is also capped by the water availability, e.g., in case of high water scarcity (during seasonal patterns).

*Wastewater generation in CWatM is represented by non-irrigation return flows, which are a function of water availability and sectoral allocation scheme, and the ratio between the consumptive and total water withdrawal.*

Another contributing factor is wastewater influent seasonality, which is associated with the collection system and dominates seasonal patterns in this manuscript. The collection of an urban runoff fraction increases the volume of collected fraction during rain events (i.e., wet season), resulting in a seasonal pattern, as shown in Figure 5 (lines 409-411) and in lines 394-397:

*Overall, the model underestimates the inflow to the Ayalon WWTP, as shown in the top panel of Error! Reference source not found., during the dry months (e.g., April to June), which is probably due to the use of annual water withdrawal inputs, that do*

*not capture seasonality. Seasonality is only captured by the 'Wastewater with urban runoff' (S2) scenario as a direct result of urban runoff collection.*

3. Line 127: "excess wastewater is discharged into pre-specified discharge locations". What exactly does this mean, and how are these specified? Are these specified discharge locations storage basins, or the surface water network itself (i.e. overflows)?

We differentiate between the potential destinations of the treated wastewater flowing out from WWTP (lines 194-196):

*The basic module has two post-treatment options: river discharge and reclamation. Direct reclamation (e.g., for irrigation purposes) is possible by using CWatM reservoirs to operate through the water demand module. This option requires data on the linkages between WWTP and reservoirs.*

It follows that river discharge, as in overflow, occurs on '*predefined overflow locations*' specifying that wastewater from WWTP can be released into the stream network and under which cases. See lines 200-203.

*If all related reservoirs are full, access water is discharged on predefined overflow locations. Discharge into streams/rivers is the default behavior if no reservoir is associated with a treatment plant. Finally, untreated wastewater is discharged if a plant's inflows exceed the plant's peak capacity (see minimally allowed HRT in section 2.2.2).*

For this assessment, we have used the exact location for the outflow and WWTP (see Table 1; lines 249-250). At a global scale, we propose a different data source (see Table 4; lines 555-559).

4. Line 140: How is the "surface area of treatment pools" estimated? Why is the estimated pool depth set to 6m? Then line 153 says the depth is set to 1m (does it differ for extensive and intensive systems; it is not clear)? Also, what is the rationale for assuming three treatment pools?

We thank the reviewer for this comment, as we have failed to include the logic behind these technical parameters. These parameters, which are the number of treatment pools (extensive system) and pool depth, are only used to calculate the water surface area and the evaporation losses. In this analysis, they are relatively low (lines 361-365):

*In the Ayalon basin case study, the largest share (68%) of the influents is being treated in the Shafdan WWTP outside of the basin of interest (i.e., Sewage*

*exported; also see Figure 2), and approximately 14% are sent to reservoirs for reclamation. The remaining share includes the discharge of treated wastewater (4%) and raw sewage (8%). Evaporation loss from WWTP is marginal (<4%) and is represented by one of the unlabeled wedges on the wastewater circle.*

The treatment systems and process representation were simplified, and the assumptions used to describe them technically are now added to the manuscript.

Regarding intensive systems (lines 159-163, and Appendix B in lines 591-606):

*Calculating the surface area of the treatment pools is different for intensive and extensive systems. **The surface area of an intensive WWTP is defined as the ratio between the plant volume and the pool depth. For that purpose, a simplified representation of WWTP treatment pool is adopted based on a clarifier design (used during both primary and secondary treatment; Pescod, 1992), and the pool depth is estimated at 6 meters (WEF, 2005; see Figure B1)***

Regarding extensive systems (lines 164-166):

*Extensive systems are modeled as **natural biological** treatment ponds, alternately filling up and treating water. **These processes consist of a relatively short anaerobic treatment in deeper ponds followed by a long-term (20-40 days) residence in facultative shallow ponds** (see Figure 1B**; also refer to Pescod, 1992**).*

We also took the opportunity to adjust our initial assumption (guided by information collected for the case study) to the general guidelines proposed by Pescod (1992; lines 173-178). The model code is also re-published.

*The surface area of each treatment pool is calculated by dividing the pool's volume by its depth (see Equation 2; Depth, **currently set to 1.5 meters, as the depth of a facultative pond; Pescod, 1992**). Each pool volume is derived by multiplying the daily capacity (VolCap) with the pool filling time. The latter is a function of the total treatment time (TreatTime) and a predefined number of treatment pools (TreatPool; currently set to **two; Pescod, 1992**). **Although evaporation losses are overall small (see Figure 4), we allow modelers to change these default technical values with their own estimates (see Appendix B).***

5. Line 173: "collected untreated wastewater is exported form the simulated region if the WWTP associated with the collection area does not exist". Please elaborate, I do not understand what is meant here.

We have rephrased the text to distinguish between two cases of interbasin wastewater transfers: (a) sending treated wastewater to other basins for reclamation purposes; (b) sending collected wastewater to treatment in other basins. The latter occurs if a service area is defined but is not associated with any WWTP in the simulated catchment (see lines 212-218).

*The module is designed to allow inter-basin transfers of wastewater or treated wastewater, yet this advanced option is not required in the case of a global model. Interbasin transfer of treated wastewater aims to account for cases in which the reclamation areas extend beyond the borders of the simulated river basin. In that case, WWTP-specific export-share parameters indicate the daily fixed percent of treated wastewater that is transferred for reclamation in other basins. The interbasin transfer of untreated wastewater represents cases in which treated wastewater collected in one basin is treated in another basin. It occurs automatically in case a defined service area is not associated with any WWTP located within the simualted basin.*

6. Line 178: How is it determined if a reservoir is accepting treated wastewater? What is the data source for reservoirs and their "associations" with wastewater treatment plants?

Data on reservoir attributes and association to WWTP and command areas based on publicly available online reports from the treatment plants associations and from a national survey was collected for the purpose of this analysis (see Table 1 lines 249-250).

At a global scale, we have proposed a scenario-based approach to assess the potential of wastewater reclamation or to construct a 'virtual' reservoir on the exact location of the WWTP to simulate direct reclamation. See Table 4 and lines 546-554:

*As advanced simulations are not pursued globally, data sources for their required variables are not sought. Reclamation and reservoirs' connections are an exception, stemming from the large impact of simulating reclamation on model performance and water resource management analysis. The reclamation rates estimated by Jones et al., (2022) can be used for that purpose. However, as it is not linked to any specific WWTP or reservoir, as required by the WTRM, it would require some pre-processing and simplifying assumptions. Some on-going efforts to identify potential wastewater reclamation for specific WWTP can support this processing (Fridman et al., 2023), yet both data sources would involve high uncertainties at the grid scale. Two other approaches could be taken to assess different reclamation scenarios, including indirect reclamation from waterbodies*

*(e.g., rivers and lakes) or simulating on-site type-4 reservoirs with command areas set as fixed buffers. Such reclamation scenarios could also explore reclamation by other non-agricultural sectors.*

The case study and scenarios are interesting. However, the model is applied and validated against observed discharge in only a single basin, which is both water scarce and already largly reliant on treated wastewater reuse. I find it therefore difficult to assess the applicability of the proposed approach in other hydrological conditions and with different levels of data availability. Being a module of a global hydrological model, I can imagine the eventual intention is to scale up this to be applicable for modelling the globe. I understand the push towards a multi-resolution modelling framework, but I currently struggle to see how/if this wastewater module would be implemented at more coarse spatial resolutions and in more data poor regions.

We have made an effort to clarify these points of criticism, aiming to show the potential of the WTRM to advance global/large-scale hydrological modeling (lines 467-470):

*The importance of including wastewater treatment and reclamation in high resolution (i.e., ~1km) hydrological modeling is also aligned with recent findings, as these models are susceptible to the effects of human activity on the water cycle and often require better representation of these processes and more precise data (Hanasaki et al., 2022).*

And since (lines 512-514): *The need for better representing wastewater treatment and reclamation in global, regional and local hydrological modeling is linked to its increasing potential as a water resource.*

We also demonstrated (as described in our comments and revised manuscript) that upscaling to a global coarser (5 arc minutes) spatial resolution is already feasible, but additional efforts could be dedicated to improving currently available data.

**Author responses to RC2**

In this manuscript, the authors introduce a new module that considers wastewater treatment and reclamation (WRTM) to the Community Water Model (CWatM). Additionally, the manuscript also provides model performance analysis of multiple scenarios with and without the new module as well as some additional wastewater balance analysis for a case study location (Ayalon basin). In general, the manuscript targets wastewater processing, which is an important aspect of research in large-scale hydrological modeling that is often overlooked. However, similar to RC1, I believe the manuscript still needs major revisions before it can be considered for publication.

We thank the reviewer for the valuable comments allowing us to improve the manuscript's readability and the quality of the analysis.

Major comments:

1. Module variables: while there are brief explanations of each variable following equations, I find some of those explanations confusing.

The text describing the module and its variables has been through a major revision (see section 2.2, lines 89-218). We included a table summarizing all the WTRM variables (Table 4; lines 555-559). Below, see our specific comments:

For example: line 109, "...a logical variable (e.g., Ddom).", what do you mean by logical? Or line 110, "...share coefficient (Cs) representing sewer connection rates and leakages", so does that mean if Cs is set at 100, the sewer system will have 100% connection with no leakage?

The term 'logical' has been replaced with 'boolean' (e.g., True/False,  1/0; lines 119-122).

*Modelling sector-specific WWTP (e.g., treatment of only industrial wastewater) is an advanced model functionality, and to-date does not fit a global application. It uses a boolean variable (e.g., $D_{Dom}$), which equales one if the treatment plant recieves a specific wastewater stream (e.g., domestic). A default value of one for both sectors is set in place, in case of missing data.*

We have rephrased to distinguish the sewer connection rate (Cs) and leakage from storm management systems to sewers (Rf x α; lines 112-118).

*WWTP service areas (or collection areas) are model input that defines the linkages between location of wastewater generation (individual grid cells, denoted by l) to wastewater treatment plants (denoted by j). Wastewater collection is also a function of the sewer connection rate ($Cs_l$; where a value of one indicates all wastewater is collected and sent to a WWTP), and can include urban runoff ($Rf_l$), due to leakage or integrtation of the urban stormwater and wastewater systems. The α coefficient defines the level of systems' integration and ranges between zero (no integration) to one (complete systems-integratuion). The total wastewater collected in all grid cells l associated with a WWTP $_j$ are registered as the treatement plant's inflow.*

Considering that this module is intended to be publicly used, it would be beneficial to both the manuscript and readers if the authors would provide a much more detailed description of the variables in Equ.1 on: (1) How can they be identified? (2) What are the ranges and units (if any) of each variable? And (3) What do specific values of each variable mean? Please consider providing additional sensitivity analysis on these parameters as well.

We have revised the text in section 2.2 (lines 89-218) to be clearer and better describe the module's processes and inputs. We added Table 4 (lines 555-559), describing all the WTRM variables, including default values. Moreover, we include a sensitivity analysis of the minimally allowed HRT in the main text (lines 398-400) and supplementary materials (lines 81-115 in the supplementary materials).

Points to adress

2. Module inputs: Table 1 has provided a summary of the model's input; however, it is unclear to the reader what the specific temporal (hourly, daily, or monthly? Timeseries or fixed value?) and spatial (gridded or vector-based?) requirements of the local datasets which hindered the possibilities of replicating this manuscript results or future applications of new model users. Additionally, it is unclear which input is critical and which is optional making it challenging to consider applying the module in other data-limited regions.

Table 1 provides an overview of the overall data requirements by CWatM and the adjustments made/new data collected to enhance model performance at a ~1 km resolution. We adjusted the structure of Table 1 better to communicate the data format, and spatial and temporal resolution.

See lines 241 -243:

*The CWatM provides global datasets at 0.5 degree and 5 arc-minutes as described in Burek et al. (2020).* *This high-resolution analysis combines global and local data sources to better represent the case-study hydrologic processes and human-hydrologic interactions (Hanasaki et al., 2022).*

and lines 246 -247:

*… A complete documentation of the dataset associated with this publication is available at https://doi.org/10.5281/zenodo.12752967.*

We added another Table 4 (lines 555-559) specifically describing the data requirements of the WTRM and indicating data availability at a global scale.

3. Model calibration: results from Table 3 suggest that while there are significant improvements when the module is applied, the simulated dry season mean flow is still substantially higher than observed data. Additionally, results in Fig B1 suggest that simulated results overestimate ET substantially across the entire calibration period (line chart) while normalized differences are also high, where the majority of the basin in spring and many locations in summer have ~70 to >100%. Thus, please consider further improving the model performance and provide additional comparisons for the validation period as well.

We thank the reviewer for pointing out this critical point. In response, we have recalibrated the models and set the following calibration scenarios:
S0: No wastewater module,
S1: Wastewater reclamation without urban runoff collection,
S2: Wastewater reclamation with urban runoff.
The results indicate that introducing the wastewater reclamation (S1, S2) and the urban runoff collection (S2) have a significant impact on model performance (section 4.1, lines 317-327).

Calibration with different features of the wastewater module also utilizes different parameters, so the calibrated model for S1 and S2 results in lower evapotranspiration flows (ET; lines 47-80 in the Supplementary material).

Further, we have enhanced the comparison of simulated ET with remote-sensing derived ET dataset by comparing with different models and expanding the discussion (section 4.1, lines 47-80 in the Supplementary material, and lines 328-336 in the revised manuscript). We have dropped the spatial comparison, as it is meaningful with some of the datasets with large to very large grid cells.

4. Model validation: considering that this paper focuses on introducing a new module, I'm struggling to see a clear comparison of model performance before and after the module is applied. It is rather difficult to distinguish the difference between simulated and observed results in Figure 3, thus, please consider providing an additional figure showing the hydrographs in a shorter period (i.e., 1 year) and displaying results from all scenarios so that the comparison is more visually clear how the new module can improve model performance. Additionally, since wastewater inflow is also a factor in evaluating model performance, please consider adding additional statistical values in Figure 5 (i.e., NSE, KGE).

Figure 3 (lines 337-339) now shows a comparison between the calibartion scenarios – for a selected year. In the supplementary materials, figures S1-S3 (lines 3-26 of the supplementary materials) show a scatter plot of observed and simulated discharge along the complete modeling period (1995-2019), for different calibartion scenarios.

Finally, table S5 (line 119 of the supplementary material) provides some metrics to compare the two scenarios to the observed WWTP inflows, as stated in lines 406-408:

***The wasterwater with urban runoff collection (S2) scenario out perfromes the scenario without wastewater collection based on multiple parametrs (showing lower bias, and higher NSE and correlation; see Table S5).***

5. Study area: similar to RC1, I find it difficult to assess the applicability of the module with just validation of one basin/one gauging station. I'd suggest the authors consider adding additional analysis on other basins where there is either less or more data available.

While applying this modelling onto additional basins is our wish, we could not allocate the resources for this task. Still, the revised manuscript indicates that a global implementation of this module is already feasible. Our argument is based on, among others, Table 4 (lines 555-559) and the revised discussion on applicability in lines 507-559.

Minor comments

1. Title, I believe the new module not only considers wastewater reclamation but also additional treatment time right, perhaps a different title that contains WRTM?

We propose a revised title including the word treatment:

*Wastewater matters: Incorporating wastewater **treatment and** reclamation into a process-based hydrological model (CWatM v1.08)*

2. Line 91, "hyper-spatial resolution", while 1km grid is relatively high-resolution compared to most global models, please consider editing this term across the manuscript since there are other small-scale models that can simulate wastewater process at much higher resolution.

Although used by Hanasaki et al. (2022) to describe a 2 km hydrological model, we have agreed to change hyper resolution to high resolution.

3. Line 87, "MODFLOW6" and line 226, "Modflow", are these two the same? If so, please be consistent across the manuscript.

All appearances are now aligned with the form MODFLOW6.

4. Figure 2, currently, the legend is rather difficult to read with all the visualization mixed in. Please consider turning all areas outside of the study area to white color in the main map to make the figure more visible.

We have remade the figure, turning the background to white over the land and blue over the sea. We have only left the OSM background for the locator map. See Figure 2, lines 288-291.

5. Figure 4, in-figure text size is too small, please consider increasing it.

We have remade all the figures and increased the font size to ensure readability.

6. Line 281-284, "... simulated evapotranspiration with a satellite derived product... the simulated monthly influent flows into the Ayalon WWTP with observed data...", please reference these figures.

Figures refs are added.

7. The "detrended values" shown in Figure 5 are not mentioned specifically anywhere in the main text, thus consider changing the text to have more linkage while adding more detailed information on the detrend technique. Additionally, there seems to be a clear increasing trend in the top panel of

Figure 5, which might be eliminated if the authors consider running an additional spin-up period to stabilize the model.

The text referring to the detrended results is in lines 401-404:

*Rain events during the wet season often result in increased inflows into the wastewater treatment plants (e.g., during December 2016 or January 2018).* ***The scenario that includes urban runoff collection (S2) can simulate these peaks, though it slightly overestimates theme, whereas no peaks are simualted for scenario S1 in which no urban runoff is collected*** *(see Figure 5 bottom panel).*

The detrending formula is now written on the y-axis label of Figure 5 bottom panel (lines 409-411).

8. All tables, table sizing, and alignments are inconsistent, please revise.

All tables' widths, alignments, font sizes, and formatting are revised for consistency.

**Author responses to RC3**

The manuscript titled "Wastewater matters: Incorporating wastewater reclamation into a process-based hydrological model (CWatM v1.08)" introduces a novel development by integrating a wastewater treatment and reclamation module into the CWatM model. This advancement is important as it incorporates human-related activities into large-scale hydrological modeling. The overall structure of the paper is sound and logically organized. However, several aspects of the methodology require further detail and clarification. Additionally, there are concerns regarding the model calibration and validation design, which could influence the results and their discussion. I recommend major revisions to address the methodology section and enhance the robustness of the model evaluation.

Detailed comments

Line 89: Please verify whether the acronym "WRTM" is correct or if it is a typographical error. The acronym for "Wastewater Treatment and Reclamation Module" should be "WTRM" rather than "WRTM". If it is a typo, please correct it throughout the manuscript.

We thank the reviewer for noticing. This was definitely a typo that slipped through. We have changed to 'WTRM' across the manuscript.

Lines 122-124: Please clarify the criteria for defining a wastewater treatment plant (WWTP) with a hydraulic retention time (HRT) ranging from 24 hours to 2 days. Would it be intensive or extensive?

Our statement was unclear. We have adapted the text as follows (lines 127-133):

*The two options are intensive and extensive treatment plants, as described in Figure 1b. Intensive treatment refers to the conventional wastewater treatment technology charcterized by low residence time and low area requirements. It usually treats water to secondary or tertiary level over less than 24 hours (Pescod, 1992). As CWatM uses a daily timestep, the intensive treatment plant's treatment period is set to one day. Any WWTP with a longer treatment period (i.e., >= 2 days) would be classified as extensive. Extensive treatment refers to natural biological systems, consist of a short primary treatment in a relatively deep anaerobic pond, followed by a longer residence time (20 -40 days) in a shallow facultative pond for secondary treatment (Pescod, 1992).*

Lines 127-130: The description here is unclear. Line 127 states that if the influent exceeds the designed capacity, the excess wastewater is discharged to a pre-specified location. This implies that the inflow is managed below designed capacity by discharging excess wastewater. However, Line 128 mentions that treatment plants allow inflows to exceed the designed capacity, which appears contradictory.

Please clarify whether "daily treatment capacity" and "designed capacity" refer to the same or different metrics.

The designed capacity can be exceeded with a potential impact on removal efficiency. These sentences, however, refer to two different modules' modes of operation. The simple mode, where data on a minimally allowed HRT is unavailable, and an advanced mode. The revised manuscript differs between these modes, for example (lines 142-146):

*According to the basic model setup,  excess wastewater beyond* the plant's daily treatment capacity *is discharged to the predefined outflow location (see Table 4). However, the model holds advanced modelling capabilities enabling higher WWTP to accept larger inflows to handle temporal fluctioations (e.g., due to significant rain events). Inflows higher that the designed capacity shortens the hydrological retention time (HRT, or residence time), resulting in less effective wastewater treatment.*

Line 136-137: Please clarify what "an increase of 25% in the operational daily capacity" is compared to? E.g., compared to HRT = 1 day?

Correct, see lines 153-155.

*For example, a minimally allowed HRT of 0.8 days implies an increase of 25% in the operational daily capacity in the case of a treatment time of 1 day.*

Lines 154-155: The term "total treatment time" is ambiguous. Is this time fixed for treating fully filled treatment pools, or does it vary based on the storage conditions? Please provide a more detailed explanation.

It has changed to a 'designed treatment time', e.g., 30 days. Lines 175-177.

*The latter is a function of the designed treatment time (TreatTime) and a predefined number of treatment pools (TreatPool; currently set to two; Pescod, 1992).*

Line 180: How are WWTPs assigned to "associated reservoirs"? For instance, must the reservoir be downstream of the WWTP, or can users assign reservoirs to WWTPs freely? Please clarify.

Reservoirs are freely associated with WWTP and provided as a model input. These, of course, should represent existing or planned conveyance systems/reclamation projects. Clarified in lines 195-197:

*This option requires data on the linkages between WWTP and reservoirs, that should represnet existing or planned water conveyance systems.*

Lines 266-267: Please clarify whether Scenario S2 was used for model calibration and if the calibrated parameters were applied across all scenarios. If so, this

approach might not be appropriate, particularly in the context of the model performance evaluation discussed in Section 4.1 (e.g., Table 3). Scenarios S0 and S1 may underperform because they were not calibrated for these specific scenarios. Please explain why separate calibration for each scenario was not performed.

We highly appreciate this comment as it precisely indicates our initial research process. Unfortunately, we have not saved these results and have had to recalibrate them. Thus, we have separately calibrated each of the calibration scenarios S0, S1, and S2 (lines 295 – 298):

*In the first scenario (S0) we disable the wastewater treatement and reclamation module. The second (S1) and third (S2) include wastewater treatment and reclamation without and with urban runoff collection, respectively. The share of urban runoff flowing into the sewers is set as a calibartion parameter in S2.*

The results were replicated relative to the former version of the manuscript and are communicated in Figure 3 (lines 337-339), Table 3 (lines 354-355), figures S1-S4 (lines 2-40 in the supplementary materials), and Figure S7 (lines 79-80 in the supplementary materials).

Figures

Figure 2: The reservoirs, canals, and other water features are difficult to distinguish due to the busy map. Consider increasing the transparency of the background map, removing it altogether, or using thicker lines to better highlight the water features.

We have remade the figure, turning the background white over the land and blue over the sea. We have only left the OSM background for the locator map. See Figure 2, lines 288-291. We also slightly modified the coloring of the reservoirs and waterbodies, and added labels and names to the reservoirs.

Figure 4: The figure is confusing, as it is unclear which text corresponds to which circle. It would be helpful to use different colors for the text associated with the circles and the text indicating different flows to improve clarity. In addition, the texts in the circles are too small to see.

We have added clear and readable labels to the wedges of the circles, along with distinct titles representing different CWatM modules. The image layout has been adjusted to enhance readability. Additionally, we opted to retain black text to avoid graphical oversaturation and maintain visual clarity. Instead, we have used different font faces, e.g., bold, normal, italic, to distinguish between circles labels, titles, and flows. See lines 378-380.

**References**

Burek, P., Satoh, Y., Kahil, T., Tang, T., Greve, P., Smilovic, M., Guillaumot, L., Zhao, F., and Wada, Y..: Development of the Community Water Model (CWatM v1.04) – a high-resolution hydrological model for global and regional assessment of integrated water resources management. *Geoscientific Model Development*, *13*(7), 3267–3298. https://doi.org/10.5194/gmd-13-3267-2020, 2020.

Ehalt Macedo, H., Lehner, B., Nicell, J., Grill, G., Li, J., Limtong, A., Shakya, R.: Distribution and characteristics of wastewater treatment plants within the global river network. *Earth System Science Data,* 14(2): 559–577. https://doi.org/10.5194/essd-14-559-2022, 2022.

Fridman, D., Kahil, T., and Wada, Y. Evaluating the global wastewater's untapped irrigation potential. *EGU General Assembly Conference Abstracts*, 2023.

Graham, D. J., Bierkens, M. F., and van Vliet, M. T. Impacts of droughts and heatwaves on river water quality worldwide. *Journal of Hydrology*, 629, 130590. https://doi.org/10.1016/j.jhydrol.2023.130590, 2024.

Hanasaki, N., Matsuda, H., Fujiwara, M., Hirabayashi, Y., Seto, S., Kanae, S., and Oki, T.: Toward hyper-resolution global hydrological models including human activities: application to Kyushu island, Japan. *Hydrology and Earth System Sciences*, *26*(8), 1953–1975. https://doi.org/10.5194/hess-26-1953-2022, 2022.

Jones, E. R., Bierkens, M. F. P., Wanders, N., Sutanudjaja, E. H., van Beek, L. P. H., and van Vliet, M. T. H.: DynQual v1.0: A high-resolution global surface water quality model. *Geoscientific Model Development Discussions*, *2022*, 1–24. https://doi.org/10.5194/gmd-2022-222, 2022.

Jones, E. R., van Vliet, M. T. H. van, Qadir, M., and Bierkens, M. F. P.: Country-level and gridded estimates of wastewater production, collection, treatment and reuse. *Earth System Science Data*, *13*(2), 237–254. https://doi.org/10.5194/essd-13-237-2021, 2021.

Pescod, M. B.: Wastewater treatment and use in agriculture - FAO irrigation and drainage paper 47, 1992.

van Vliet, M. T. H., Jones, E. R., Flörke, M., Franssen, W. H. P., Hanasaki, N., Wada, Y., and Yearsley, J. R.: Global water scarcity including surface water quality and expansions of clean water technologies. *Env. Res. Lett.*, 16: 024020. https://doi.org/10.1088/1748-9326/abbfc3, 2021.

Wang, C., Song, J., Nunes, L. M., Zhao, H., Wang, P., Liang, Z., Arp, H. P. H., Li, G. and Xing, B. Global microplastic fiber pollution from domestic laundary. *Journal of Hazardous Materials*, 577, 135290. https://doi.org/10.1016/j.jhazmat.2024.135290, 2024.

Water Environment Federation (WEF): Clarifier design, Second edition. Manual of Practice No. FD-8. Water Environment Federation, 2005.

---

## Author Response (AR2)

**Authors' responses to reviewer comments**

We thank the reviewers and the editor for their constructive feedback on our manuscript. Following are our responses to their comments. Blue indicates our responses, *italics* is *used for citations from the manuscript*, and **bold shows implemented changes**. All the lines, figures, or tables mentioned below refer to the revised track changes manuscript.

**Topic editor**

**Public justification (visible to the public if the article is accepted and published)**:
Dear Authors,

Thanks again for your detailed response to the reviewers' initial comments, and for your patience in enduring our delay in issuing this decision (things predictably slowed down over the holidays). I asked the reviewers to consider your response to their comments and revised manuscript. As you will see, the referees generally felt that you addressed their initial concerns, though some concerns remain. Please provide individual responses to their latest round of comments, and I would appreciate it if you would pay particular attention to addressing reviewer 3's concerns, as I had the same concerns myself regarding the reproducibility of the study's repository/results and the quality of figures.

In addition to the reviewers' comments, here are some additional comments from me, some of which have overlaps with the reviewers' comments.

1) Software Development. With regard to software development, it is difficult to easily discern the specific modifications to CWatM that were undertaken to conduct this study. The authors are encouraged to discuss (and display through a figure if possible) more clearly the modifications and how they fit into the existing CWATM model structure (and its presumably numerous modules). I feel that this is actually rather important for readers to be able to figure out how they might go about extending what you have done (e.g., to a global scale, or to add new types of reuse (which I mention below).

We have revised Figure 1 (Lines 101-104) to highlight the processes added to CWatM by the WTRM. These are now shown in a different fill color. We have also reorganized that plot to enhance readability.

We further revised the Code and Data Availability to provide readers with relevant tools to support using CWatM and the module and data associated with this manuscript

(Lines 638-644).

Please also refer to the response to Ref 3.

2) The quality of some figures is a bit concerning to me--both the image/file quality but also the figure organization/appearance. For example, Figures 3 and 4 are a bit disorganized and difficult to read (i.e., see labels, discern trends). Please see Reviewer 3's suggestions for enhancing figure clarity.
Following this comment and the comment from Ref. 3, we have revised all our plots, particularly Figures 3 and 4.

In Figure 3:

- we have trimmed some of the months to focus on the relevant data,
- we have also trimmed the y-axis so the differences between different scenarios is better captured in the graphs,
- We have improved the graphic language and included labels to indicate better the role of the different components in the graph (Line 350-353).

In Figure 4:

- we have simplified the graphics, including just the wastewater circle (instead of three), and added some straight arrows and numbered labels to account for the interaction with other model components (Line 407-409),
- describing texts are also updated (Lines 372-393),
- a complementary reservoir water circle is added in Figure S4.

3) What parts of the landscape of water reuse options that currently exist are modeled here versus excluded? For example, readers may wonder, is direct potable reuse handled? How about indirect groundwater injection? It would be good for readers to know where this model fits, so they can quickly identify its applicability to their specific research interests. In a sense, this also relates to some initial concerns I had about the lack of discussion of limitations. Please help readers understand what they can and cannot use this model for in a variety of reuse contexts.
The WTRM interacts with CWatM, allowing the simulation of diverse reuse options. In line 203, we have updated some examples: *The basic module has two post-treatment options: river discharge and* **reuse**. *Direct* **reuse** *(e.g., for irrigation*, **industrial, and potable uses***) is possible using the CWatM reservoirs and water demand routines.*

This assessment focuses primarily on wastewater reuse for irrigation and, to some extent, reuse for livestock purposes (Lines 304-307):

*"The second (S1) and third (S2) include wastewater treatment and reuse without and with urban runoff collection, respectively. The share of urban runoff flowing into the sewers is set as a calibration parameter in S2. **In this case study, we defined sectoral water allocations to limit wastewater reuse to irrigation, with limited use for livestock purposes".***

This is also mentioned in Lines 519-522:

*"Combined with the source-sector abstraction fraction, the modeling of the Ayalon basin has limited the use of treated wastewater for irrigation and livestock to a smaller extent. Other existing uses, like urban landscaping or cooling of thermal powerplants, were **excluded**, as data to incorporate those uses into our module was unavailable".*

We have also added a paragraph suggesting potential uses, which are partially demonstrated in this analysis (Lines 508-510):

**"Reservoirs allow storing and transferring treated wastewater, and reusing it in relevant irrigation districts (i.e., by utilizing the CWatM command areas feature). Leakage from reservoirs into groundwater aquifers (see Figure S4) can be used to simulate groundwater recharge with treated wastewater."**

Although some additional options (e.g., treated wastewater injection into the aquifer) can be included (e.g., by defining negative values of groundwater abstraction in MODFLOW6), there are currently no variables/processes set in CWatM to represent these processes. Therefore, we do not include them in the current study.

4) Regarding terminology, your revision addressed a lot of my initial concerns, but I do think there are some areas still where defining terms could help. For example, how does wastewater reclamation differ from wastewater reuse? Many readers will not understand whether/how you distinguish between these terms.

We have used wastewater reuse and reclamation interchangeably, as they have a similar meaning. Wastewater reclamation includes collecting, treating, and reusing wastewater, whereas reuse only refers to the last process. We accept that using these two terms interchangeably may confuse the reader, and we replaced the word reclamation with reuse. As such, the paper title has been amended to: *Wastewater matters: Incorporating wastewater treatment and **reuse** into a process-based hydrological model (CWatM v1.08)*.

Many thanks for your submission and for your revision efforts.

Thanks,

Tom

**Ref 1:**

Thanks to the authors for the re-submission, and for their efforts to improve the manuscript.

Overall, I am mostly satisfied with the revisions (although some corrections to typographical errors are still required). Please find below my response/additional comments on the manuscript, organised with respect to my original reviewer report (i.e. RC1).

1. The authors have done a good job in better describing the three stages of the workflow (i.e. pre-treatment, treatment and post-treatment) and have provided additional clarity on some of the more ambiguous terms previously used (e.g. "user defined collection areas"). The distinction between the "simple" and "advanced" options for the module are now also more clear, and the additional discussion is also helpful. Note that the reference in lines 525-526 should be Jones et al., (2023) [Jones, E. R., Bierkens, M. F. P., Wanders, N., Sutanudjaja, E. H., van Beek, L. P. H., and van Vliet, M. T. H.: DynQual v1.0: a high-resolution global surface water quality model, Geosci. Model Dev., 16, 4481–4500, https://doi.org/10.5194/gmd-16-4481-2023, 2023.]. Also note that the pre-print version of this paper is cited in different places in the manuscript, which should also be updated to the 2023 published paper.

We thank the reviewer for the positive assessment of our revision. The reference and citations throughout the papers were updated.

2. Thanks for the clarification; I interpret this as wastewater generation being simulated by CWatM-WQ with a daily timestep, but read into WTRM as input as an annual or monthly average – is that correct? Also not sure I fully understand why there is not a seasonality component in the estimates of inflow associated with water withdrawals, in addition to S2 which includes a seasonality component as a result of urban runoff. Can you briefly explain?

Wastewater collection is the sum of the collected wastewater generation and urban runoff.

Wastewater generation collection lacks seasonality due to the dataset used in this assessment.

We have better distinguished the newly introduced features of WTRM from those of CWatM, as seen in the revised Figure 1 (Lines 101-104). Generated wastewater is non-irrigation return flows, simulated by CWatM daily. The volume of non-irrigation return

flows is a function of water withdrawals for domestic and industrial uses (which depend on water demand and availability) and return-flow fraction.

1/ Water demand and return-flow fraction are model inputs in CWatM, estimated annually or monthly, following a similar approach to the PCR-GLOBWB's (Wada et al., 2011a, 2011b). Annual/monthly water withdrawal is divided within CWatM by the number of days in a calendar year/month.

In this analysis, we have constructed a national dataset based on annual municipal data from the Israeli Bureau of Statistics.

2/ Water availability can limit water withdrawal, thus reducing non-irrigation withdrawal.

In this analysis, as desalination acts as a backstop water source, water availability is not a limiting factor for wastewater generation.

Seasonality in inflows into WWTP originates, in this assessment, from the urban runoff component.

Based on the discussion above, it could be acceptable that wastewater generation in this assessment is based on annual data and lacks seasonality. However, wastewater collection includes, to some extent, urban runoff, which has a seasonal component.

References:

Wada, Y., van Beek, L. P. H., and Bierkens, M. F. P. 2011a. Modelling global water stress of the recent past: on the relative importance of trends in water demand and climate variability. *Hydrol. Earth Syst. Sci.,* 15, 3785–3808, https://doi.org/10.5194/hess-15-3785-2011, 2011.

Wada, Y., van Beek, L. P. H., Viviroli, D., Dürr, H. H., Weingartner, R., and Bierkens, M. F. P. 2011b. Global monthly water stress: 2. Water demand and severity of water stress. *Water Resources Research,* 47(7): W07518, doi:10.1029/2010WR009792.

3. OK, that is now more clear.

4. OK.

5. I find this aspect still somewhat unclear. Perhaps the manuscript could benefit from a more detailed description of how "wastewater treatment service areas" are defined (comment 1). The authors response to this comment makes it sound like these service areas are defined based on river basin borders (and then the model checks if there is a WWTP in that area); and therefore not delineating these zones based on the precise location of wastewater treatment plants? If that is the case, what then happens if multiple WWTPs are located in the same river basin? Apologies if I misunderstand here.

We have now included another map in the Supplementary (Figure S10), demonstrating the two main collection areas (aka service areas) of the *Ayalon* and *Shafdan* WWTPs.

To make the notion of collection areas more understandable, we have included additional explanations in lines 119-121:

*"WWTP service areas (or collection areas) are model input that defines the linkages between the location of wastewater generation (individual grid cells, denoted by l) and wastewater treatment plants (denoted by j)**, namely that the wastewater from all grid cells in a collection area are treated in the associated WWTP (see Figure S10)"**.*

Further, we elaborate on how we have created the data (Lines 559-563):

*"Two additional challenges are indicated in Table 4, associated with the treatment days and service (wastewater collection) area**. In this study, we rely on a national dataset associating municipalities with WWTPs (see Figure S10; INRA, 2016), yet this data is not available for most countries. Instead, following** Ehalt Macedo et al. (2022), the **wastewater collection areas** can be traced back from the WWTP to serve the nearest, most likely upstream, population centers".*

In the supplementary of Ehalt Macedo et al. (2022) the authors outline three different approaches to estimate the number of people served by a WWTP (P. 4-8 of the supplementary of Ehalt Macedo et al., 2022). Two of the approaches (A2 & A3) rely on the assumption that the grid cells that are closer to a WWTP are more likely to be served by it, and they propose to set a buffer (e.g., optimal size found to be 11 km) around each WWTP to find its population. Based on this approach, it would be possible to trace back the most likely service areas for each WWTP. However, we do not find the description of that approach, and these ideas are within this paper's scope.

References:

Ehalt Macedo, H., Lehner, B., Nicell, J., Grill, G., Li, J., Limtong, A., and Shakya, R. 2022. Distribution and characteristics of wastewater treatment plants within the global river network, *Earth Syst. Sci. Data*, 14, 559–577, https://doi.org/10.5194/essd-14-559-2022, 2022.

6. OK.

7. For clarification, I certainly don't doubt the importance of including wastewater treatment and reclamation in (high-resolution) hydrological models. My comment was more related to the fact that the model has only been applied to a single basin (albeit with various scenarios) that is both water scarce and reliant on wastewater; and therefore find it somewhat difficult to assess the model performance in different

hydrological/socioeconomic settings. But perhaps this comes later, once the additional efforts to upscale the approach to global scale have been made.

We thank the reviewer for the clarification. We also appreciated that comment, as the reproducibility component was not sufficiently developed in the first version of this manuscript. We hope our additional explanations clarify the upscaling potential of our module, and we are planning additional case study applications for the future.

**Ref 2:**

**No comments**

**Ref 3 (poor reproducibility, poor presentation quality):**

The authors have addressed my technical comments. However, the overall quality of the paper still does not meet the standards of GMD. The reproducibility of the study is not clear, considering there is no detailed instructions in the repository.

We thank the reviewer for these insights and suggestions, which helped us to improve the manuscript further. To allow reproducibility, we have taken the following steps:

1/ added into the Code and Data Availability additional references to relevant tools that support readers in using CWatM and the newly developed module in this manuscript, which reads in lines 638-644 as follows:

*The CWatM code is provided through a GitHub repository (https://github.com/iiasa/CWatM; last accessed: 15 February 2025), and the model version used for this study (CWatM-Israel v1.06.1) is provided via* **https://doi.org/10.5281/zenodo.13990296 (Fridman, 2024; last accessed: 25/10/2024). Documentation and tutorials of CWatM are available at https://cwatm.iiasa.ac.at/ (last accessed: 15 February 2025).** *The input data used for this publication, including model settings and initial conditions files, can be downloaded from  https://zenodo.org/doi/10.5281/zenodo.13990451 (***Fridman et al., 2025;** *last accessed:* **26/02/2025**)*.

*2/ We have updated the dataset associated with this manuscript, by –*

- *revising the preview text,*
- *and adding a readme file with details on the dataset content and association between settings files and simulated scenarios (e.g., specific graphs and tables in the manuscript).*

The authors should consider making significant improvements to the presentation quality, particularly in the figures and tables, to enhance clarity. **The issues highlighted below are not isolated to a single figure or table but are indicative of broader presentation quality across the paper. Authors should carefully review and revise all figures and tables to improve the overall quality and readability of the manuscript.**

We have reviewed all figures and tables in the manuscript and included changes to Figure 1, Table 1, Figure 3, and Figure 4 (please see also responses to the editor's comments).

Figure 1:

1. **Add more details to clarify the relationship between the new WTRM and CWatM**. Clearly distinguish what constitutes the original CWatM and what are the new additions in the WTRM.

**2**. Ensure consistent font sizes for the same hierarchical elements. For instance, the font size describing processes like "intensive" and "extensive" systems should match the font size of the text within the Figure 1A box.

**3**. Use standard notations such as "(A)" and "(B)" to label subfigures for better readability and consistency.

**4**. Clearly label both "active pond" and "inactive pond" to distinguish between these components in the extensive treatment system.

**5**. Avoid using duplicate number labels within the same plot. For example, the number "1" appears twice for "Pre-treatment" and "Intensive," which could confuse readers.

Figure 1 is replaced with a revised version (Lines 101-104) where CWatM's original features and newly introduced features are separated using fill color; labels were controlled for size and rearranged, and repetitive numbering.

Table 1:

**1**. Improve the structure of the table by making the separation between global and local datasets more apparent. Consider centering and bolding the headers for these two sections.

**2**. Standardize temporal resolution terminology (e.g., "annual" vs. "by year") for consistency.

**3**. Break down rows with multiple datasets into separate rows for better clarity. For instance:

**a. Separate the "wastewater treatment plants database" into three distinct rows.**

**b. Apply a similar approach to the "groundwater basin and aquifers" row.**

In Table 1 (Line 258), we have broken lines that were aggregated together; we have divided the spatial and temporal resolutions into two columns and standardized the terms used to describe each.

On some occasions, we have left some items as an aggregate, e.g., *Wastewater attributes and technical data*; in that case, we add a brief description of the included variables (usually 2-3 variables) into the data source and comments column, e.g., *Attributes include wastewater treatment levels, years of operation*.

Figure 3:

1. **Significant improvements are required to enhance readability. For example, it is difficult to distinguish the color difference between the "No wastewater" and "Wastewater without urban runoff" lines.**

2. **Revisit the overall design to ensure that line colors, labels, and legends are clearly distinguishable.**

Figure 3 (Lines 350-353) was revised to enhance its visual appearance and readability. Particularly,

- the 'No wastewater' line color is now darker,
- we have trimmed some of the months to focus on the relevant data,
- we have also trimmed the y-axis so the differences between different scenarios are better captured in the graphs,
- and we have improved the graphic language and included labels to better indicate the role of the different components in the graph.